# Union of Intersections (UoI) for Interpretable Data Driven Discovery and Prediction

**Kristofer E. Bouchard**[*]    **Alejandro F. Bujan**[†]    **Farbod Roosta-Khorasani**[‡]

**Shashanka Ubaru**[§]    **Prabhat**[¶]    **Antoine M. Snijders**[||]    **Jian-Hua Mao**[||]

**Edward F. Chang**[**]    **Michael W. Mahoney**[‡]    **Sharmodeep Bhattacharyya**[††]

## Abstract

The increasing size and complexity of scientific data could dramatically enhance discovery and prediction for basic scientific applications. Realizing this potential, however, requires novel statistical analysis methods that are both interpretable and predictive. We introduce Union of Intersections (UoI), a flexible, modular, and scalable framework for enhanced model selection and estimation. Methods based on UoI perform model selection and model estimation through intersection and union operations, respectively. We show that UoI-based methods achieve low-variance and nearly unbiased estimation of a small number of interpretable features, while maintaining high-quality prediction accuracy. We perform extensive numerical investigation to evaluate a UoI algorithm ($UoI_{Lasso}$) on synthetic and real data. In doing so, we demonstrate the extraction of interpretable functional networks from human electrophysiology recordings as well as accurate prediction of phenotypes from genotype-phenotype data with reduced features. We also show (with the $UoI_{L1Logistic}$ and $UoI_{CUR}$ variants of the basic framework) improved prediction parsimony for classification and matrix factorization on several benchmark biomedical data sets. These results suggest that methods based on the UoI framework could improve interpretation and prediction in data-driven discovery across scientific fields.

## 1   Introduction

A central goal of data-driven science is to identify a small number of features (i.e., predictor variables; $X$ in Fig. 1(a)) that generate a response variable of interest ($y$ in Fig. 1(a)) and then to estimate the relative contributions of these features as the parameters in the generative process relating the predictor variables to the response variable (Fig. 1(a)). A common characteristic of many modern massive data sets is that they have a large number of features (i.e., high-dimensional data), while

---

[*]Biological Systems and Engineering Division, LBNL. `kebouchard@lbl.gov`

[†]Redwood Center, UC Berkeley. `afbujan@gmail.com`

[‡]ICSI and Department of Statistics, UC Berkeley. `{farbod,mmahoney}@icsi.berkeley.edu`

[§]Department of Computer Science and Engineering, University of Minnesota. `ubaru001@umn.edu`

[¶]NERSC, LBNL. `prabhat@lbl.gov`

[||]Biological Systems and Engineering Division, LBNL. `{AMSnijders,jhmao}@lbl.gov`

[**]Department of Neurological Surgery, UC San Francisco. `Edward.Chang@ucsf.edu`

[††]Department of Statistics, Oregon State University. `bhattash@science.oregonstate.edu`

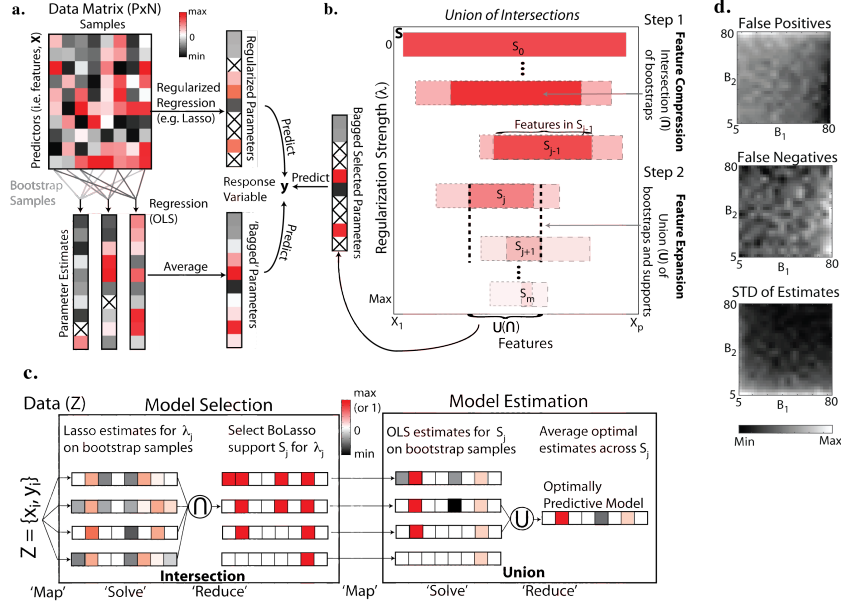

Figure 1: **The basic UoI framework.** (a) Schematic of regularization and ensemble methods for regression. (b) Schematic of the Union of Intersections (UoI) framework. (c) A data-distributed version of the $UoI_{Lasso}$ algorithm. (d) Dependence of false positive, false negatives, and estimation variability on number of bootstraps in selection ($B_1$) and estimation ($B_2$) modules.

also exhibiting a high degree of sparsity and/or redundancy [2, 19, 11]. That is, while formally high-dimensional, most of the useful information in the data features for tasks such as reconstruction, regression, and classification can be restricted or compressed into a much smaller number of important features. In regression and classification, it is common to employ sparsity-inducing regularization to attempt to achieve simultaneously two related but quite different goals: to identify the features important for prediction (i.e., model selection) and to estimate the associated model parameters (i.e., model estimation) [2, 19]. For example, the Lasso algorithm in linear regression uses $L_1$-regularization to penalize the total magnitude of model parameters, and this often results in feature compression by setting some parameters exactly to zero [18] (See Fig. 1(a), pure white elements in right-hand vectors, emphasized by $\times$). It is well known that this type of regularization implies a prior assumption about the distribution of the parameter (e.g., $L_1$-regularization implicitly assumes a Laplacian prior distribution) [12]. However, strong sparsity-inducing regularization, which is common when there are many more potential features than data samples (i.e., the so-called small $n/p$ regime) can severely hinder the *interpretation* of model parameters (Fig. 1(a), indicated by less saturated colors between top and bottom vectors on right hand side). For example, while sparsity may be achieved, incorrect features may be chosen and parameters estimates may be biased. In addition, it can impede model selection and estimation when the true model distribution deviates from the assumed distribution [2, 10]. This may not matter for prediction quality, but it clearly has negative consequences for *interpretability*, an admittedly not completely-well-defined property of algorithms that is crucial in many scientific applications [9]. In this context, *interpretability* reflects the degree to which an algorithm returns a small number of physically meaningful features with unbiased and low variance estimates of their contributions.

On the other hand, another common characteristic of many state of the art methods is to combine several related models for a given task. In statistical data analysis, this is often formalized by so-called

ensemble methods, which improve prediction accuracy by combining parameter estimates [12]. In particular, by combining several different models, ensemble methods often include more features to predict the response variables, and thus the number of data features is expanded relative to the individuals in the ensemble. For example, estimating an ensemble of model parameters by randomly resampling the data many times (e.g., bootstrapping) and then averaging the parameter estimates (e.g., bagging) can yield improved prediction accuracy by reducing estimation variability [8, 12] (See Fig. 1(a), bottom). However, by averaging estimates from a large ensemble, this process often results in many non-zero parameters, which can hinder interpretability and the identification of the true model support (compare top and bottom vectors on right hand side of Fig. 1(a)). Taken together, these observations suggest that explicit and more precise control of feature compression and expansion may result in an algorithm with improved interpretative and predictive properties.

In this paper, we introduce Union of Intersections (UoI), a flexible, modular, and scalable framework to enhance both the identification of features (model selection) as well as the estimation of the contributions of these features (model estimation). We have found that the UoI framework permits us to explore the interpretability-predictivity trade-off space, without imposing an explicit prior on the model distribution, and without formulating a non-convex problem, thereby often leading to improved interpretability and prediction. Ideally, data analysis methods in many scientific applications should be selective (only features that influence the response variable are selected), accurate (estimated parameters in the model are as close to the true value as possible), predictive (allowing prediction of the response variable), stable (e.g., the variability of the estimated parameters is small), and scalable (able to return an answer in a reasonable amount of time on very large data sets) [17, 2, 15, 10]. We show empirically that UoI-based methods can simultaneously achieve these goals, results supported by preliminary theory. We primarily demonstrate the power of UoI-based methods in the context of sparse linear regression ($UoI_{Lasso}$), as it is the canonical statistical/machine learning problem, it is theoretically tractable, and it is widely used in virtually every field of scientific inquiry. However, our framework is very general, and we demonstrate this by extending UoI to classification ($UoI_{L1Logistic}$) and matrix factorization ($UoI_{CUR}$) problems. While our main focus is on neuroscience (broadly speaking) applications, our results also highlight the power of UoI across a broad range of synthetic and real scientific data sets.[1]

## 2   Union of Intersections (UoI)

For concreteness, we consider an application of UoI in the context of the linear regression. Specifically, we consider the problem of estimating the parameters $\beta \in \mathbb{R}^p$ that map a $p$-dimensional vector of predictor variables $x \in \mathbb{R}^p$ to the observation variable $y \in \mathbb{R}$, when there are $n$ paired samples of $x$ and $y$ corrupted by i.i.d Gausian noise:

$$y = \beta^T x + \varepsilon, \tag{1}$$

where $\varepsilon \overset{iid}{\sim} N(0, \sigma^2)$ for each sample. When the true $\beta$ is thought to be sparse (i.e., in the $L_0$-norm sense), then an estimate of $\beta$ (call it $\hat{\beta}$) can be found by solving a constrained optimization problem of the form:

$$\hat{\beta} \in \text{argmin}_{\beta \in \mathbb{R}^p} \sum_{i=1}^{n} (y_i - \beta x_i)^2 + \lambda R(\beta). \tag{2}$$

Here, $R(\beta)$ is a regularization term that typically penalizes the overall magnitude of the parameter vector $\beta$ (e.g., $\mathbb{R}(\beta) = \|\beta\|_1$ is the target of the Lasso algorithm).

**The Basic UoI Framework.** The key mathematical idea underlying UoI is to perform model selection through intersection (compressive) operations and model estimation through union (expansive) operations, in that order. This is schematized in Fig. 1(b), which plots a hypothetical range of selected

features ($x_1 : x_p$, abscissa) for different values of the regularization parameter ($\lambda$, ordinate). See [4] for a more detailed description. In particular, UoI first performs feature compression (Fig. 1(b), Step 1) through intersection operations (intersection of supports across bootstrap samples) to construct a family ($S$) of candidate model supports (Fig. 1(b), e.g., $S_{j-1}$, opaque red region is intersection of abutting pink regions). UoI then performs feature expansion (Fig. 1(b), Step 2) through a union of (potentially) different model supports: for each bootstrap sample, the best model estimates (across different supports) is chosen, and then a new model is generated by averaging the estimates (i.e., taking the union) across bootstrap samples (Fig. 1(b), dashed vertical black line indicates the union of features from $S_j$ and $S_{j+1}$). Both feature compression and expansion are performed across all regularization strengths. In UoI, feature compression via intersections and feature expansion via unions are balanced to maximize prediction accuracy of the sparsely estimated model parameters for the response variable $y$.

**Innovations in Union of Intersections.** UoI has three central innovations: (1) calculate model supports ($S_j$) using an intersection operation for a range of regularization parameters (increases in $\lambda$ shrink all values $\hat{\beta}$ towards 0), efficiently constructing a family of potential model supports $\{S : S_j \in S_{j-k}, \text{for } k \text{ sufficiently large}\}$; (2) use a novel form of *model averaging* in the union step to directly optimize prediction accuracy (this can be thought of as a hybrid of bagging [8] and boosting [16]); and (3) combine pure model selection using an intersection operation with model selection/estimation using a union operation in that order (which controls both false negatives and false positives in model selection). Together, these innovations often lead to better selection, estimation and prediction accuracy. Importantly, this is done without explicitly imposing a prior on the distribution of parameter values, and without formulating a non-convex optimization problem.

**The $UoI_{Lasso}$ Algorithm.** Since the basic UoI framework, as described in Fig. 1(c), has two main computational modules—one for model selection, and one for model estimation—UoI is a framework into which many existing algorithms can be inserted. Here, for simplicity, we primarily demonstrate UoI in the context of linear regression in the $UoI_{Lasso}$ algorithm, although we also apply it to classification with the $UoI_{L1Logistic}$ algorithm as well as matrix factorization with the $UoI_{CUR}$ algorithm. $UoI_{Lasso}$ expands on the BoLasso method for the model selection module [1], and it performs a novel *model averaging* in the estimation module based on averaging ordinary least squares (OLS) estimates with potentially different model supports. $UoI_{Lasso}$ (and UoI in general) has a high degree of natural algorithmic parallelism that we have exploited in a distributed Python-MPI implementation. (Fig. 1(c) schematizes a simplified distributed implementation of the algorithm; see [4] for more details.) This parallelized $UoI_{Lasso}$ algorithm uses distribution of bootstrap data samples and regularization parameters (in *Map*) for independent computations involving convex optimizations (Lasso and OLS, in *Solve*), and it then combines results (in *Reduce*) with intersection operations (model selection module) and union operations (model estimation module). By solving independent convex optimization problems (e.g., Lasso, OLS) with distributed data resampling, our $UoI_{Lasso}$ algorithm efficiently constructs a family of model supports, and it then averages nearly unbiased model estimates, potentially with different supports, to maximize prediction accuracy while minimizing the number of features to aid interpretability.

## 3 Results

### 3.1 Methods

All numerical results used 100 random sub-samplings with replacement of 80-10-10 cross-validation to estimate model parameters (80%), choose optimal meta-parameters (e.g., $\lambda$, 10%), and determine prediction quality (10%). Below, $\beta$ denotes the values of the true model parameters, $\hat{\beta}$ denotes the estimated values of the model parameters from some algorithm (e.g., $UoI_{Lasso}$), $S_\beta$ is the support of

the true model (i.e., the set of non-zero parameter indices), and $S_{\hat{\beta}}$ is the support of the estimated model. We calculated several metrics of model selection, model estimation, and prediction accuracy. (1) Selection accuracy (set overlap): $1 - \frac{|S_{\hat{\beta}} \Delta S_\beta|}{|S_{\hat{\beta}}|_0 + |S_\beta|_0}$, where, $\Delta$ is the symmetric set difference operator. This metric ranges in $[0, 1]$, taking a value of 0 if $S_\beta$ and $S_{\hat{\beta}}$ have no elements in common, and taking a value of 1 if and only if they are identical. (2) Estimation error (r.m.s): $\sqrt{\frac{1}{p} \sum \left(\beta_i - \hat{\beta}_i\right)^2}$. (3) Estimation variability (parameter variance): $E[\hat{\beta}^2] - (E[\hat{\beta}])^2$. (4) Prediction accuracy ($R^2$): $\frac{\sum (y_i - \hat{y}_i)^2}{\sum (y_i - E[y])^2}$. (5) Prediction parsimony (BIC): $n \log(\frac{1}{n-1} \sum_{i=1}^n (y_i - \hat{y}_i)^2) + \|\hat{\beta}\|_0 \log(n)$. For the experimental data, as the true model size is unknown, the selection ratio ($\frac{\|\hat{\beta}\|_0}{p}$) is a measure of the overall size of the estimated model relative to the total number of parameters. For the classification task using $UoI_{L1Logistic}$, BIC was calculated as: $-2 \log \ell + S_{\hat{\beta}} \log N$, where $\ell$ is the log-likelihood on the validation set. For the matrix factorization task using $UoI_{CUR}$, reconstruction accuracy was the Frobenius norm of the difference between the data matrix $A$ and the low-rank approximation matrix $A'$ constructed from $A(:, c)$, the reduced column matrix of A: $\|A - A'\|_F$, where $c$ is the set of $k$ selected columns.

## 3.2 Model Selection and Stability: Explicit Control of False Positives, False Negatives, and Estimate Stability

Due to the form of the basic UoI framework, we can control both false negative and false positive discoveries, as well as the stability of the estimates. For any regularized regression method like in (2), a decrease in the penalization parameter ($\lambda$) tends to increase the number of false positives, and an increase in $\lambda$ tends to increase false negatives. Preliminary analysis of the UoI framework shows that, for false positives, a large number of bootstrap resamples in the intersection step ($B_1$) produces an increase in the probability of getting no false positive discoveries, while an increase in the number of bootstraps in the union step ($B_2$) leads to a decrease in the probability of getting no false positives. Conversely, for false negatives, a large number of bootstrap resamples in the union step ($B_2$) produces an increase in the probability of no false negative discoveries, while an increase in the number of bootstraps in the intersection step ($B_1$) leads to a decrease in the probability of no false negatives. Also, a large number of bootstrap samples in union step ($B_2$) gives a more stable estimate. These properties were confirmed numerically for $UoI_{Lasso}$ and are displayed in Fig. 1(d), which plots the average normalized false negatives, false positives, and standard deviation of model estimates from running $UoI_{Lasso}$, with ranges of $B_1$ and $B_2$ on four different models. These results are supported by preliminary theoretical analysis of a variant of $UoI_{Lasso}$ (see [4]). Thus, the relative values of $B_1$ and $B_2$ express the fundamental balance between the two basic operations of intersection (which compresses the feature space) and union (which expands the feature space). Model selection through intersection often excludes true parameters (i.e., false negatives), and, conversely, model estimation using unions often includes erroneous parameters (i.e., false positives). By using stochastic resampling, combined with model selection through intersections, followed by model estimation through unions, UoI permits us to mitigate the feature inclusion/exclusion inherent in either operation. Essentially, the limitations of selection by intersection are counteracted by the union of estimates, and vice versa.

## 3.3 $UoI_{Lasso}$ has Superior Performance on Simulated Data Sets

To explore the performance of the $UoI_{Lasso}$ algorithm, we have performed extensive numerical investigations on simulated data sets, where we can control key properties of the data. There are a large number of algorithms available for linear regression, and we picked some of the most popular algorithms (e.g., Lasso), as well as more uncommon, but more powerful algorithms (e.g., SCAD, a non-convex method). Specifically, we compared $UoI_{Lasso}$ to five other model selection/estimation

**Figure 2**

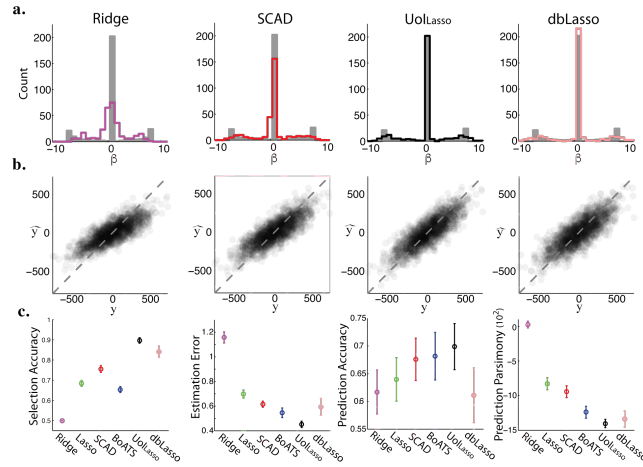

Figure 2: **Range of observed results, in comparison with existing algorithms.** (a) True $\beta$ distribution (grey histograms) and estimated values (colored lines). (b) Scatter plot of true and estimated values of observation variable on held-out samples. (c) Metrics of algorithm performance.

methods: Ridge, Lasso, SCAD, BoATS, and debiased Lasso [12, 18, 10, 5, 3, 13]. Note that BoATS and debiased Lasso are both two-stage methods. We examined performance of these algorithms across a variety of underlying distributions of model parameters, degrees of sparsity, and noise levels. Across all algorithms examined, we found that $UoI_{Lasso}$ (Fig. 2, black) generally resulted in very high selection accuracy (Fig. 2(c), right) with parameter estimates with low error (Fig. 2(c), center-right), leading to the best prediction accuracy (Fig. 2(c), center-left) and prediction parsimony (Fig. 2(c), left). In addition, it was very robust to differences in underlying parameter distribution, degree of sparsity, and magnitude of noise. (See [4] for more details.)

### 3.4 $UoI_{Lasso}$ in Neuroscience: Sparse Functional Networks from Human Neural Recordings and Parsimonious Prediction from Genetic and Phenotypic Data

We sought to determine if the enhanced selection and estimation properties of $UoI_{Lasso}$ also improved its utility as a tool for data-driven discovery in complex, diverse neuroscience data sets. Neurobiology seeks to understand the brain across multiple spatio-temporal scales, from molecules-to-minds. We first tackled the problem of graph formation from multi-electrode ($p = 86$ electrodes) neural recordings taken directly from the surface of the human brain during speech production ($n = 45$ trials each). See [7] for details. That is, the goal was to construct sparse neuroscientifically-meaningful graphs for further downstream analysis. To estimate functional connectivity, we calculated partial correlation graphs. The model was estimated independently for each electrode, and we compared the results of graphs estimated by $UoI_{Lasso}$ to the graphs estimated by SCAD. In Fig. 3(a)-(b), we display the networks derived from recordings during the production of /b/ while speaking /ba/. We found that the $UoI_{Lasso}$ network (Fig. 3(a)) was much sparser than the SCAD network (Fig. 3(b)). Furthermore, the network extracted by $UoI_{Lasso}$ contained electrodes in the lip (dorsal vSMC), jaw (central vSMC), and larynx (ventral vSMC) regions, accurately reflecting the articulators engaged in the production of /b/ (Fig. 3(c)) [7]. The SCAD network (Fig. 3(d)) did not have any of these properties. This highlights the improved power of $UoI_{Lasso}$ to extract sparse graphs with functionally meaningful features relative to even some non-convex methods.

We calculated connectivity graphs during the production of 9 consonant-vowel syllables. Fig. 3(e) displays a summary of prediction accuracy for $UoI_{Lasso}$ networks (red) and SCAD networks (black)

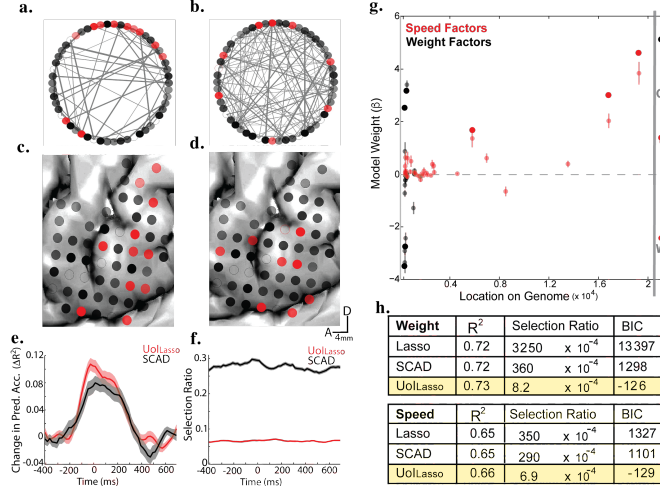

Figure 3: **Application of UoI to neuroscience and genetics data.** (a)-(f): Functional connectivity networks from ECoG recordings during speech production. (g)-(h): Parsimonious prediction of complex phenotypes form genotype and phenotype data.

as a function of time. The average relative prediction accuracy (compared to baseline times) for the $UoI_{Lasso}$ network was generally greater during the time of peak phoneme encoding [T = -100:200] compared to the SCAD network. Fig. 3(f) plots the time course of the parameter selection ratio for the $UoI_{Lasso}$ network (red) and SCAD network (black). The $UoI_{Lasso}$ network was consistently $\sim 5\times$ sparser than the SCAD network. These results demonstrate that $UoI_{Lasso}$ extracts sparser graphs from noisy neural signals with a modest increase in prediction accuracy compared to SCAD.

We next investigated whether $UoI_{Lasso}$ would improve the identification of a small number of highly predictive features from genotype-phenotype data. To do so, we analyzed data from $n = 365$ mice (173 female, 192 male) that are part of the genetically diverse Collaborative Cross cohort. We analyzed single-nucleotide polymorphisms (SNPs) from across the entire genome of each mouse ($p = 11,563$ SNPs). For each animal, we measured two continuous, quantitative phenotypes: weight and behavioral performance on the rotorod task (see [14] for details). We focused on predicting these phenotypes from a small number of geneotype-phenotype features. We found that $UoI_{Lasso}$ identified and estimated a small number of features that were sufficient to explain large amounts of variability in these complex behavioral and physiological phenotypes. Fig. 3(g) displays the non-zero values estimated for the different features (e.g., location of loci on the genome) contributing to the weight (black) and speed (red) phenotype. Here, non-opaque points correspond to the mean $\pm$ s.d. across cross-validation samples, while the opaque points are the medians. Importantly, for both speed and weight phenotypes, we confirmed that several identified predictor features had been reported in the literature, though by different studies, e.g., genes coding for Kif1b, Rrm2b/Ubr5, and Dloc2. (See [4] for more details.) Accurate prediction of phenotypic variability with a small number of factors was a unique property of models found by $UoI_{Lasso}$. For both weight and rotorod performance, models fit by $UoI_{Lasso}$ had marginally increased prediction accuracy compared to other methods ($+1\%$), but they did so with far fewer parameters (lower selection ratios). This results in prediction parsimony (BIC) that was several orders of magnitude better (Fig. 3(h)). Together, these results demonstrate that $UoI_{Lasso}$ can identify a small number of genetic/physiological factors that are highly predictive of complex physiological and behavioral phenotypes.

**a.**

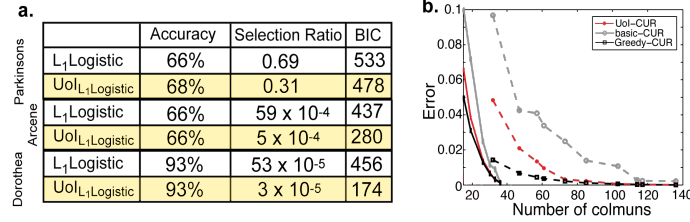

| | Accuracy | Selection Ratio | BIC |
|---|---|---|---|
| $L_1$Logistic | 66% | 0.69 | 533 |
| UoI$_{L_1 Logistic}$ | 68% | 0.31 | 478 |
| $L_1$Logistic | 66% | 59 x 10$^{-4}$ | 437 |
| UoI$_{L_1 Logistic}$ | 66% | 5 x 10$^{-4}$ | 280 |
| $L_1$Logistic | 93% | 53 x 10$^{-5}$ | 456 |
| UoI$_{L_1 Logistic}$ | 93% | 3 x 10$^{-5}$ | 174 |

Figure 4: **Extension of UoI to classification and matrix decomposition.** (a) UoI for classification ($UoI_{L1Logistic}$). (b) UoI for matrix decomposition ($UoI_{CUR}$); solid and dashed lines are for PAH and dashed SORCH data sets, respectively.

### 3.5 $UoI_{L1Logistic}$ and $UoI_{CUR}$: Application of UoI to Classification and Matrix Decomposition

As noted, UoI is is a framework into which other methods can be inserted. While we have primarily demonstrated UoI in the context of linear regression, it is much more general than that. To illustrate this, we implemented a classification algorithm ($UoI_{L1Logistic}$) and matrix decomposition algorithm ($UoI_{CUR}$), and we compared them to the base methods on several data sets (see [4] for details). In classification, UoI resulted in either equal or improved prediction accuracy with 2x-10x fewer parameters for a variety of biomedical classification tasks (Fig. 4(a)). For matrix decomposition (in this case, column subset selection), for a given dimensionality, UoI resulted in reconstruction errors that were consistently lower than the base method (BasicCUR), and quickly approached an unscalable greedy algorithm (GreedyCUR) for two genetics data sets (Fig. 4(b)). In both cases, UoI improved the prediction parsimony relative to the base (classification or decomposition) method.

## 4 Discussion

UoI-based methods leverage stochastic data resampling and a range of sparsity-inducing regularization parameters/dimensions to build families of potential features, and they then average nearly unbiased parameter estimates of selected features to maximize predictive accuracy. Thus, UoI separates model selection with intersection operations from model estimation with union operations: the limitations of selection by intersection are counteracted by the union of estimates, and vice versa. Stochastic data resampling can be a viewed as a perturbation of the data, and UoI efficiently identifies and robustly estimates features that are stable to these perturbations. A unique property of UoI-based methods is the ability to control both false positives and false negatives. Initial theoretical work (see [4]) shows that increasing the number of bootstraps in the selection module ($B_1$) increases the amount of feature compression (primary controller of false positives), while increasing the number of bootstraps in the estimation module ($B_2$) increases feature expansion (primary controller of false negatives), and we observe this empirically. Thus, neither should be too large, and their relative values express the balance between feature compression and expansion. This tension is seen in many places in machine learning and data analysis: local nearest neighbor methods vs. global latent factor models; local spectral methods that tend to expand due to their diffusion-based properties vs. flow-based methods that tend to contract; and sparse $L_1$ vs. dense $L_2$ penalties/priors more generally. Interestingly, an analogous balance of compressive and expansive forces contributes to neural leaning algorithms based on Hebbian synaptic plasticity [6]. Our results highlight how revisiting popular methods in light of new data science demands can lead to still further-improved methods, and they suggest several directions for theoretical and empirical work.

## Footnotes

[1]More details, including both empirical and theoretical results, are in the associated technical report [4].

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
