[Supplementary Material · uoi_tr.pdf]

# Union of Intersections (UoI) for Interpretable Data Driven Discovery and Prediction

Kristofer E. Bouchard[*]     Alejandro F. Bujan[†]     Farbod Roosta-Khorasani[‡]

Shashanka Ubaru[§]     Prabhat[¶]     Antoine M. Snijders[‖]     Jian-Hua Mao[**]

Edward F. Chang[††]     Michael W. Mahoney[‡‡]     Sharmodeep Bhattacharyya[§§]

## Abstract

The increasing size and complexity of scientific data could dramatically enhance discovery and prediction for basic scientific applications. Realizing this potential, however, requires novel statistical analysis methods that are both interpretable and predictive. We introduce Union of Intersections (UoI), a flexible, modular, and scalable framework for enhanced model selection and estimation. Methods based on UoI perform model selection and model estimation through intersection and union operations, respectively. We show that UoI-based methods achieve low-variance and nearly unbiased estimation of a small number of interpretable features, while maintaining high-quality prediction accuracy. We perform extensive numerical investigation to evaluate a UoI algorithm ($UoI_{Lasso}$) on synthetic and real data. In doing so, we demonstrate the extraction of interpretable functional networks from human electrophysiology recordings as well as accurate prediction of phenotypes from genotype-phenotype data with reduced features. We also show (with the $UoI_{L1Logistic}$ and $UoI_{CUR}$ variants of the basic framework) improved prediction parsimony for classification and matrix factorization on several benchmark biomedical

[*]Biological Systems and Engineering Division, Lawrence Berkeley National Laboratory; Computational Research Division, Lawrence Berkeley National Laboratory; Redwood Center for Theoretical Neuroscience, University of California at Berkeley; Kavli Institute for Fundamental Neuroscience, University of California at San Francisco; Email: kebouchard@lbl.gov

[†]Redwood Center for Theoretical Neuroscience, University of California at Berkeley; Email: afbujan@gmail.com

[‡]International Computer Science Institute; Department of Statistics, University of California at Berkeley; Email: farbod@icsi.berkeley.edu

[§]Department of Computer Science and Engineering, University of Minnesota-Twin Cities; Email: ubaru001@umn.edu

[¶]National Energy Research Super Computing, Lawrence Berkeley National Laboratory; Email: prabhat@lbl.gov

[‖]Biological Systems and Engineering Division, Lawrence Berkeley National Laboratory; Email: AMSnijders@lbl.gov

[**]Biological Systems and Engineering Division, Lawrence Berkeley National Laboratory; Email: jhmao@lbl.gov

[††]UCSF Epilepsy Center, University of California at San Francisco; Department of Neurological Surgery, University of California at San Francisco; Email: Edward.Chang@ucsf.edu

[‡‡]International Computer Science Institute; Department of Statistics, University of California at Berkeley; Email: mmahoney@stat.berkeley.edu

[§§]Department of Statistics, Oregon State University; Email: bhattash@science.oregonstate.edu

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

# A  Additional Material

In this appendix section, we will provide additional information about the UoI method. We will start in Section A.1 with an extended introduction for scientific data. Then, in Section A.2, we provide pseudo-code for $UoI_{Lasso}$; in Section A.3, we discuss scaling issues; in Section A.4, we describe preliminary theoretical analysis of Union of Intersections; in Section A.5, we discuss expanded results for the simulated data example; in Section A.6, we discuss simulated data across different parameter distributions and levels of sparsity; in Section A.7, we discuss simulated data across different noise magnitudes; in Section A.8, we discuss $UoI_{L1Logistic}$ for classification problems; and in Section A.9, we discuss $UoI_{CUR}$ for applying matrix decompositions to genetics data. We conclude in Section A.10 with a brief additional discussion and conclusion.

## A.1  Extended Introduction for Scientific Examples

An important aspect of the use of machine learning and data analysis techniques in scientific applications—as opposed to internet, social media, and related applications—is that scientific researchers often implicitly or explicitly interpret the output of their data analysis tools as reflecting the true state of nature. For example, in neuroscience, one often wants to understand how neural activity (e.g., action potentials, calcium transients, cortical field potentials, etc.) is mapped to features of the external world (e.g., sounds or movement), or to features of the brain itself (e.g., the activity of other brain areas) [15]. A common approach to this is to formulate the mapping as a parametric model and then estimate the model parameters from noisy data. In addition to providing predictive capabilities, such model parameters can also provide insight into neural representations, functional connectivity, and population dynamics [36, 39, 6]. Indeed, this insight into the underlying neuroscience is typically at least as important as the predictive quality of the model. Likewise, in molecular biology and medicine, recent advances have allowed for the proliferation of low-cost whole-genome mapping, paving the way for large-scale genome wide association studies (GWAS) [42]. The relationship between genetic variations and observed phenotypes can be estimated from a parametric model [19, 12, 43]. Here, researchers may be interested in methods that allow for the identification of low-penetrance genes that are present at high frequency in a population, as these are likely the major genetic components associated with predisposition to disease risk and other physiological and behavioral phenotypes [28, 24]. Because the molecules encoded by the genes are often used to guide future experiments or drug development, identifying a small number of genetic factors that are highly predictive is critical to accelerate basic discovery and targets for next generation therapeutics. These and other examples [14] illustrate that the prediction-versus-interpretation data analysis needs of the scientific community are not well aligned with the needs of the Internet and social media industries that are forcing functions for the development of many machine learning and data analysis methods.

## A.2 Pseudo-code for the $UoI_{Lasso}$ Algorithm

Here, we provide pseudo-code for a $UoI_{Lasso}$ algorithm.

**Algorithm:** $UoI_{Lasso}$ Input: data $(X,Y) \in \mathbb{R}^{n \times (p+1)}$;
vector of regularization parameters $\lambda \in \mathbb{R}^q$;
number of bootstraps $B_1$ and $B_2$;

**% Model Selection**
for k=1 to $B_1$ do
    Generate bootstrap sample $T^k = (X_T^k, Y_T^k)$
    for $\lambda_j \in \lambda$ do
        Compute Lasso estimate $^j\hat{\beta}^k$ from $T^k$
        Compute support $S_j^k = \{i\}$ s.t. $^j\hat{\beta}_i^k \neq 0$
    end for
end for
for j=1 to q do
    Compute BoLasso support for $\lambda_j : S_j = \bigcap_{k=1}^{B_1} S_j^k$
end for

**% Model Estimation**
for k=1 to $B_2$ do
    Generate bootstrap samples for cross-validation:
    training $T^k = (X_T^k, Y_T^k)$
    evaluation $E^k = (X_E^k, Y_E^k)$
    for j=1 to q do
        Compute OLS estimate $\hat{\beta}_{S_j}^k$ from $T^k$
        Compute loss on $E^k : L(\hat{\beta}_{S_j}^k, E^k)$
    end for
    Compute best model for each bootstrap sample:
    $\hat{\beta}_S^k = \underset{\hat{\beta}_{S_j}^k}{\operatorname{argmin}} L(\hat{\beta}_{S_j}^k, E^k)$
end for
Compute bagged model estimate $\hat{\beta}^* = \frac{1}{B_2} \sum_{k=1}^{B_2} \hat{\beta}_S^k$
Return: $\hat{\beta}^*$

As described in the main text, UoI is a framework that includes performing complementary unions and intersections of a basic underlying method. When applied to $L_1$-regularized $L_2$ regression, we obtain the $UoI_{Lasso}$ algorithm. This pseudo-code implements a serial version of the $UoI_{Lasso}$ algorithm. The algorithm uses the BoLasso method for the model selection through the intersection module, and bagged ordinary least squares regression for the the estimation through the union module. This algorithm is described serially, but, as schematized in Fig. 1(c), this algorithm has a great deal of natural parallelism, involving independent calculations across different bootstrap samples ($B_1$ and $B_2$) and values of the regularization parameter ($\lambda$), which occur for both the selection and estimation modules. Of course, further algorithmic parallelization can be achieved by distributing the computations required for solving the Lasso and OLS convex-optimization steps, using, e.g., the Alternating Directions Method of Multiplies (ADMM). Even for linear regression, other methods could be used for the selection module (e.g., SCAD, stability selection, etc.), though keeping the estimation module as an un-regularized method should be maintained. When other base algorithms are used, e.g., a logistic classifier or a CUR matrix decomposition, then other variants of the basic UoI framework, such as $UoI_{L1Logistic}$ and $UoI_{CUR}$ (that are described below and in the main text), are obtained.

## A.3  Scaling of the $UoI_{Lasso}$ Algorithm

$UoI_{Lasso}$ (and UoI in general) has a high degree of natural algorithmic parallelism that we have exploited in a distributed Python-MPI implementation. (See Section 3.1 also.) To assess the scalability of the algorithm, we carried out a series of scaling computations with data sets of different sizes; and, in Fig. 5, we present a summary of the performance between a serial and a distributed implementation of $UoI_{Lasso}$. We used the computational runtime and the input-output (IO) time as performance indicators. We used artificially generated data sets with sizes that ranged from 400 bytes to 40 gigabytes, therefore spanning 5 orders of magnitude in size, and we performed computations on a supercomputer (NERSC at LBNL), as described in Section 3.1. Fig. 5(a) shows the computational runtime for the distributed and the serial program compared across data set sizes, as indicated by the gray scale color and the legend. All data points in Fig 5(a) lie well above the identity line shown as a diagonal gray dashed line, which indicates that runtime was in all cases notably lower for the distributed version of $UoI_{Lasso}$. This improvement in runtime increased with the data set size: the best-fit line to the (log-log) data had a slope of 1.19 and y-intercept of 2.13. This indicates a general improvement of approximately two-orders of magnitude (y-intercept), and that improvements get better for larger data sets (slope greater than 1). The computational runtime is also shown in Fig. 5(b) with pink and gray bars, and in addition this plot includes the IO time shown with red and black bars for the parallel and serial versions of $UoI_{Lasso}$, respectively. We found that even though IO time was slightly larger in the parallel version for small data sets, IO operations

Figure 5: **Efficient and scalable implementation of $UoI_{Lasso}$ on distributed computing systems.** (a) Comparison of computational runtime between serial and distributed $UoI_{Lasso}$, as a function of the data set size. Gray-scale: size of the design matrix in bytes (legend). Circular markers: computational runtime of serial $UoI_{Lasso}$ was estimated using a single iteration. Square markers: actual data points. Red line: linear fit to the actual (log-log) data (squares). Grey dashed line is unity. (b) Computational runtime and data IO time for serial (grey/black) and distributed (pink/red) $UoI_{Lasso}$ as a function of data set size (x-axis). Hatched bars: estimated runtime time using a single iteration.

benefited from the distributed implementation when large datasets were analyzed. Overall, these results show a good scalability, in parallel settings, of the basic UoI framework, illustrating its potential applicability to very large-scale data sets.

## A.4   Theory

In section 3.2, we make certain statements regarding the control of false negative and false positive discoveries in $UoI_{Lasso}$ method and the relationship of the control of false positive and false negative discoveries in $UoI_{Lasso}$ method with the bootstrap parameters $B_1$ and $B_2$. The bootstrap parameter $B_1$ is used in the *model selection* or intersection step of the $UoI_{Lasso}$ algorithm. The *model selection* step derives from the BoLasso algorithm [1]. As shown in Lemma 6 in section A.4.5, for correct model selection with high probability, we should have $B_1 \to \infty$ but at a rate slower than $\log n$ for BoLasso. But for theoretical analysis of the model estimation step of $UoI_{Lasso}$ algorithm, we need separate control on the false positives (regression coefficients falsely estimated as non-zero) and false negatives (regression coefficients falsely estimated as zero) of the recovered support. In Lemma 6, the result gives a bound on total error in support recovery, but not separate control on false positives and false negatives. For this reason, in the theoretical analysis of UoI methods, in stead of using BoLasso in the model selection step, we use stability selection [33], as there are some better

theoretical properties available for the stability selection method. We give the pesudo-code of the modified algorithm, $UoI_{Lasso}^{stable}$, in section A.4.1. Since, the bootstrap parameter for *model selection* step of $UoI_{Lasso}$, $B_1$, is not a parameter of the algorithm $UoI_{Lasso}^{stable}$, the main theoretical results proved for $UoI_{Lasso}^{stable}$ establish relationships between the false positive and false negative control and $B_2$, the bootstrap parameter for the *model estimation* step.

Let us consider that we have data $(Y_1, \boldsymbol{X}_1), \ldots, (Y_n, \boldsymbol{X}_n)$ with univariate response variable $Y_i$ and $p$-dimensional predictor variable $\boldsymbol{X}_i$ for each sample, $i \in \{1, \ldots, n\}$. The vectors $(Y_i, \boldsymbol{X}_i)$ are assumed independent with common distribution in $\mathbb{R}^{p+1}$ for each $i \in \{1, \ldots, n\}$. Consider the linear regression model for the data

$$\boldsymbol{Y} = \boldsymbol{X}\boldsymbol{\beta} + \boldsymbol{\varepsilon} \tag{3}$$

where, $\boldsymbol{Y} = (Y_1, \ldots, Y_n)$, $\boldsymbol{X}$ is the $n \times p$ random design matrix of explanatory variables and $\boldsymbol{\varepsilon} = (\varepsilon_1, \ldots, \varepsilon_n)$ are random noise terms with $\boldsymbol{\varepsilon} \stackrel{iid}{\sim} N(0, \sigma^2 \mathbf{I}_n)$ and $\boldsymbol{X}_i$ are orthogonal to $\varepsilon_i$ for each $i \in \{1, \ldots, n\}$, that is, $\mathbb{E}(X_{ij}\varepsilon_i) = 0$ for $j = 1, \ldots, p$. Also, the design matrix has the property $\sum_{i=1}^n \boldsymbol{X}_{ij}^2 = 1$ for all $j = 1, \ldots, p$. Let $S$ be the set of non-zero coefficients of $\boldsymbol{\beta}$ with $|S| = s$; $N$ be the set of zero coefficients of $\boldsymbol{\beta}$ with $|N| = p - s$. We consider that $s$ is constant and $p$ can be function of $n$, the sample size.

Consider the lasso regression problem with regularization parameter $\lambda > 0$, as minimizing the following optimization function with respect to $\boldsymbol{\beta}$,

$$L(\boldsymbol{\beta}, \lambda) = ||\boldsymbol{Y} - \boldsymbol{X}\boldsymbol{\beta}||_2^2 + \lambda ||\boldsymbol{\beta}||_1 \tag{4}$$

### A.4.1 $UoI_{Lasso}$ Algorithm with Stability Selection

Here we present a preliminary theoretical analysis of Union of Intersections for Lasso based regression. The $UoI_{Lasso}$ algorithm is presented in section A.2. The algorithm analyzed in this section differs slightly from the $UoI_{Lasso}$ algorithm in that it uses the stability selection method of Meinshausen and Buhlmann (2010) [33] instead of BoLasso in the model selection step. This was for tractability, as the current analytical results for stability selection are more amenable to theoretical analysis in the UoI framework. A brief review of the results on BoLasso is given in section A.4.5. As noted in [33], there is a great deal of similarity between stability selection and BoLasso. The stability selection method proposed in [33] has two hyperparameters, $\alpha \in \mathbb{R}_+$ (where, $\mathbb{R}_+ = \{x \in \mathbb{R} | x > 0\}$) and $\pi_{thr} \in [0, 1]$ (for details, see [33]). As mentioned in [33], the stability selection algorithm is similar to the BoLasso algorithm for $\pi_{thr} = 1$.

Here, we provide pseudo-code for $UoI_{Lasso}^{stable}$ algorithm.

**Algorithm: $UoI_{Lasso}^{stable}$** Input: data $(X, Y) \in \mathbb{R}^{n \times (p+1)}$;
vector of regularization parameters $\boldsymbol{\lambda} \in \mathbb{R}_+^q$;

Stability selection hyperparameters $\alpha \in \mathbb{R}_+$ and $\pi_{thr} \in (0,1)$;

number of bootstraps $B_2$;

**% Model Selection**

for j=1 to q do

Compute stability selection support for $\lambda_j : S_j$, by running the stability selection algorithm from [33] with hyperparameters $(\alpha, \pi_{thr}, \lambda_j)$ on data $(\boldsymbol{Y}, \boldsymbol{X})$.

end for

**% Model Estimation**

for k=1 to $B_2$ do

Generate bootstrap samples for cross-validation:

training $T^k = ((X_T^k)_{n_1 \times p}, (Y_T^k)_{n_1 \times 1})$

evaluation $E^k = ((X_E^k)_{n_2 \times p}, (Y_E^k)_{n_2 \times 1})$

for j=1 to q do

Compute OLS estimate $\hat{\beta}_{S_j}^k$ from $T^k$

Compute loss on $E^k : L(\hat{\beta}_{S_j}^k, \lambda_j)$

end for

Compute best model for each bootstrap sample:

$\hat{\beta}_S^k = \underset{\hat{\beta}_{S_j}^k : j=1,\dots p}{\text{argmin}} \; L(\hat{\beta}_{S_j}^k, \lambda_j)$

end for

Compute bagged model estimate $\hat{\beta}_{UoI}^{stable} = \frac{1}{B_2} \sum_{k=1}^{B_2} \hat{\beta}_S^k$

Return: $\hat{\beta}_{UoI}^{stable}$

Based on the estimate $\hat{\beta}_{UoI}^{stable}$, we define the *support* of the coefficient estimate as the set of non-zero coefficients of $\hat{\beta}_{UoI}^{stable}$. We call the *support set* to be $\hat{S}_{B_2}^{UoI} \equiv \{j \in \{1, \dots, p\} | (\beta_{UoI}^{stable})_j \neq 0\}$. The UoI$_{\text{Lasso}}^{\text{stable}}$ algorithm works in two parts. In the first part, that is the *model selection* step, a support set for the coefficients is obtained for each $\lambda \in \boldsymbol{\lambda}$ (where, $\boldsymbol{\lambda}$ is the set of regularization parameters) based on stability selection algorithm. In the second part, that is the *model estimation* step, a least squares estimate is obtained using the support set obtained from model selection step for $B_2$ bootstrap samples of the data and for each $\lambda \in \boldsymbol{\lambda}$. For each bootstrap sample, the *best* coefficient estimate corresponding to the regularization parameter $\lambda$ with minimum prediction error is recorded. A new estimate of the regression coefficients is obtained by taking the average of the *best* estimates obtained for each bootstrap sample. Thus the support of the regression coefficient estimate ultimately obtained is found after the union of the support of several stability selection

based estimates.

Another set of notations is needed for theoretical analysis of the $\text{UoI}_{\text{Lasso}}^{\text{stable}}$ algorithm. For any $A \subseteq \{1, \ldots, p\}$, define the sub-design and sub-Gram matrices as

$$X_A \equiv (X^j, j \in A)_{n \times |A|}, \quad \Sigma_A = X_A^T X_A \tag{5}$$

where, $X^j$ is the $j$-th column of design matrix or the observations corresponding to $j$-th predictor variable.

We consider several assumptions on the set-up to prove the theoretical results for the $\text{UoI}_{\text{Lasso}}^{\text{stable}}$ algorithm. The assumptions (A1)-(A2) are required for properly defining the linear model. The assumption (A1) states the condition on the random design matrix set up. It states the assumption of independence of data and independence between explanatory variables and error variables. It also gives condition on the distribution of the response and explanatory variables. The assumption (A2) specifies the linear model and homoscedasticity assumptions. Assumption (A3) gives condition on the covariance matrix of the design or explanatory variables. The bound on the ratio of largest and smallest eigenvalues of sub-matrices of covariance matrix of explanatory variables given in (A3) is required in proving the results on model selection as well as model estimation step. Lastly, the assumption (A4) states the restriction on the size of the bootstrap training and validation samples in the model estimation step. The condition (A4) is needed so that both estimation of parameters using training sample and estimation of regularization parameters using validation sample have nice large sample properties.

**Assumption:**

(A1) The vectors $(\boldsymbol{Y}_i, \boldsymbol{X}_i)$ are assumed independent with common (unknown) distribution in $\mathbb{R}^{p+1}$. The cumulant generating functions $\mathbb{E}(\exp(z||X||_2^2))$ and $\mathbb{E}(\exp(zY^2))$ are finite for some $z > 0$. Also, $Y_i - X_i\boldsymbol{\beta}$ is orthogonal to $X_i$, that is, $\mathbb{E}(X_{ij}\boldsymbol{\varepsilon}_i) = 0$ for $i = 1, \ldots, n$ and $j = 1, \ldots, p$, where, $\boldsymbol{\varepsilon}_i = Y_i - X_i\boldsymbol{\beta}$.

(A2) $\mathbb{E}(\boldsymbol{Y}|\boldsymbol{X}) = \boldsymbol{X}^T\boldsymbol{w}$ and $\text{Var}(\boldsymbol{Y}|\boldsymbol{X}) = \sigma^2$ a.s. for some $\boldsymbol{w} \in \mathbb{R}^p$ and $\sigma \in \mathbb{R}_+$.

(A3) We consider the sparse Reisz condition (SRC) as given in [33]. Let us consider that there exists functions $c_{min} : \{1, \ldots, p\} \to \mathbb{R}_+$ and $c_{max} : \{1, \ldots, p\} \to \mathbb{R}_+$, such that for $d \in \mathbb{Z}$, $d > 0$, we have,

$$c_{min}(d) \leq \min_{|A| \leq d} \phi_{min}(\Sigma_A) \leq \max_{|A| \leq d} \phi_{max}(\Sigma_A) \leq c_{max}(d) \tag{6}$$

where, $\phi_{min}(M)$ and $\phi_{max}(M)$ are the minimum and maximum eigenvalues of a matrix $M$. In that case, we assume that there exists some constant $C > 1$ and some constant $\kappa \geq 9$, such that,

$$\frac{c_{max}(Cs^2)}{c_{min}^{3/2}(Cs^2)} < \sqrt{C}\kappa$$

with probability, $p_s$, where, $p_s \to 1$, as $n, p \to \infty$.

(A4) The number of training and validation samples in the model selection step, $n_1$ and $n_2$ should follow the relation that $n_1 = c_1 n$ and $n_2 = c_2 n$, where, $0 < c_1, c_2 < 1$ are constants (not dependent on $p$).

Under the assumptions (A1)-(A4) on the linear model setup, we prove the following result on model selection and model estimation accuracy of the UoI procedure. Note that, we denote, $(a \wedge b) := \min\{a, b\}$.

**Theorem 1.** *Consider the model in* (3) *and the assumptions (A1)-(A4) is satisfied.*

(a) **Model selection step:** *Consider $\alpha$ given by $\alpha^2 = \nu c_{min}(m)/m$, for any $\nu \in ((7/\kappa)^2, 1/\sqrt{2})$, and $m = Cs^2$, for some constant $C > 1$. Let $a_n$ be a sequence with $a_n \to \infty$ for $n \to \infty$. Let $\lambda_{min} = 2\sigma(\sqrt{2C}s + 1)\sqrt{\log(p \wedge a_n)/n}$ and assume that $p > 10$ and $s \geq 7$. Then, there exists some $\delta = \delta_s \in (0, 1)$ such that for all $\pi_{thr} \geq 1 - \delta$, there exists a set $\Omega_1$ with $\mathrm{P}(\Omega_1) \geq p_s(1 - 5/(p \wedge a_n))$, such that for data set $(Y_i, \boldsymbol{X}_i)_{i=1}^n$ arising from such a set,*

$$N \cap \hat{S}_\lambda^{stable} = \emptyset, \tag{7}$$

*where $\hat{S}_\lambda^{stable} =$ support selected by stability selection with $\lambda \geq \lambda_{min}$. On the same set $\Omega_1$,*

$$(S \backslash S_{small;\lambda}) \subseteq \hat{S}_\lambda^{stable} \tag{8}$$

*where $S_{small;\lambda} = \{k : |\beta_k| \leq 0.3(Cs)^{3/2}\lambda\}$.*

(b) **Model estimation step:** *For a set $\Omega$ such that $\mathrm{P}(\Omega) \geq \min\left\{1 - (1 - p_W)^{B_2}, (p_W)^{B_2}\right\}$, where, $p_W = p_s^2(1 - 5/(p \wedge a_n))$ and for data $(\boldsymbol{Y}_i, \boldsymbol{X}_i)_{i=1}^n$ as element of the set $\Omega$, we have,*

$$N \cap \hat{S}_{B_2}^{UoI} = \emptyset \tag{9}$$

*where $\hat{S}_{B_2}^{UoI} =$ support selected after model estimation step. On the same set $\Omega$,*

$$(S \backslash S_{small;\lambda_{min}}) \subseteq \hat{S}_{B_2}^{UoI} \tag{10}$$

*where, $S_{small;\lambda_{min}} = \{k : |\beta_k| \leq 0.3(Cs)^{3/2}\lambda_{min}\}$.*

(c) **Estimation Accuracy:** *On the same set $\Omega$ as in (b), for data $(\boldsymbol{Y}_i, \boldsymbol{X}_i)_{i=1}^n$ as element of the set $\Omega$, we have,*

$$\left\|X(\hat{\beta}_{UoI}^{stable} - \beta)\right\|^2 \leq C_1 \sigma^2 \frac{\log p}{n} \tag{11}$$

*for some constant $C_1 > 0$.*

**Comments:** The following observations can be made from the above Theorem 1.

(a) **Explanation.** In part (a) and (b) of the Theorem, equations (7) and (9), the results imply that no noise variables are selected. In equation (8) and (10), the results imply that all variables with sufficiently large regression coefficient are selected.

(a) **Control of false positive discoveries.** The control of false positive discoveries is achieved both in the *model selection* and *model estimation* steps. The probability of having no false positives in the model selection step, $p_s \left(1 - \frac{5}{p \wedge a_n}\right)$, which tends to one as $p, n \to \infty$. Although it is not explicitly stated in the Theorem, it becomes apparent from the proof that the union step that the probability of having no false positives to $p_W^{B_2}$, where, $p_W = p_s^2 \left(1 - 5/(p \wedge a_n)\right)$, which tend to one as $n, p \to \infty$. Note that, the probability of having no false positives decreases in the model estimation step to $p_W^{B_2}$ from $p_s \left(1 - \frac{5}{p \wedge a_n}\right)$ in model selection step for $B_2 \geq 1$. Also, $p_W^{B_2} \to 1$ if $B_2 \to \infty$ slow enough.

(b) **Control of false negative discoveries.** The control of false negative discoveries is also achieved both in the *model selection* and *model estimation* steps. The maximum size of false negatives in the model selection step is $S_{small;\lambda} = \{k : |\beta_k| \leq 0.3(Cs)^{3/2}\lambda\}$ for each $\lambda \in \boldsymbol{\lambda}$ and such false negatives occur with probability $p_s \left(1 - \frac{5}{p \wedge a_n}\right)$, which tends to one as $p, n \to \infty$. Although it is not explicitly stated in the Theorem, the model estimation step both decreases the probability of having false negatives as well as reduces the size of the false negatives. The maximum size of false negatives in the model estimation step is $S_{small;\lambda_{min}} = \{k : |\beta_k| \leq 0.3(Cs)^{3/2}\lambda_{min}\}$ and such false negatives occur with probability $(1 - (1 - p_W)^{B_2})$. So, the maximum size of false negatives after the model estimation step, $S_{small;\lambda_{min}}$, becomes smaller than the size $S_{small;\lambda}$ obtained after model selection step for $\lambda \geq \lambda_{min}$. The number of non-zero parameter values estimated as zero becomes small with probability $(1 - (1 - p_W)^{B_2})$, where, $p_W = p_s^2(1 - 5/(p \wedge a_n))$, which tends to one as $n, p \to \infty$. Thus, a large number of bootstrap resamples in the model estimation step ($B_2$) produce an increase in the probability of having small false negative discoveries. Also, the probability increases in the model estimation step to $(1 - (1 - p_W)^{B_2})$ from $p_s \left(1 - \frac{5}{p \wedge a_n}\right)$ in model selection step for large enough $B_2$. Thus the UoI$_{\text{Lasso}}^{\text{stable}}$ algorithm improves upon the stability selection results.

(c) **Model selection in the whole UoI operation.** After the model estimation step, we have correct model selection with high probability for the entire UoI procedure if both the probabilities $(1 - (1 - p_W)^{B_2})$ and $p_W^{B_2}$ are large.

(d) **Estimation Accuracy.** The estimation accuracy of the UoI$_{\text{Lasso}}^{\text{stable}}$ estimators occurs at the same rate as Lasso but with probability as $\min \left\{1 - (1 - p_W)^{B_2}, (p_W)^{B_2}\right\}$.

### A.4.2 Proof of Theorem 1

For the sake of clarity, we redefine several notations. Let us consider the estimated coefficient parameter and the set of selected variables from stability selection (or model selection step) for any regularization parameter $\lambda \in \mathbb{R}_+$ to be $\hat{\boldsymbol{\beta}}_\lambda$ and $\hat{S}_\lambda$ respectively. So, we can see that by the notation in the algorithm $\text{UoI}_{\text{Lasso}}^{\text{stable}}$, $S_j \equiv \hat{S}_{\lambda_j}$, where, $\lambda_j$ was one of the penalization parameters used as input in algorithm $\text{UoI}_{\text{Lasso}}^{\text{stable}}$. Let us also consider that for any penalization parameter $\lambda \in \mathbb{R}_+$, $s_\lambda := |\hat{S}_\lambda \cap S|$ and $n_\lambda := |\hat{S}_\lambda \cap S^C|$. In the rest of the section, we also change some of the notations from $\text{UoI}_{\text{Lasso}}^{\text{stable}}$ algorithm. In the $k^{th}$ ($k = 1, \ldots, B_2$) iteration of model estimation step in $\text{UoI}_{\text{Lasso}}^{\text{stable}}$, we redefine the training data based on bootstrap to be $(\boldsymbol{Z}_{n_1 \times 1}, \boldsymbol{W}_{n_1 \times p})$ (which was denoted as $(\boldsymbol{Y}_T^k, \boldsymbol{X}_T^k)$ in the $\text{UoI}_{\text{Lasso}}^{\text{stable}}$ algorithm) and we redefine the validation data based on bootstrap to be $((\boldsymbol{Z}_0)_{n_2 \times 1}, (\boldsymbol{W}_0)_{n_2 \times p})$ (which was denoted as $(\boldsymbol{Y}_E^k, \boldsymbol{X}_E^k)$ in the $\text{UoI}_{\text{Lasso}}^{\text{stable}}$ algorithm). Note that, the training and the validation data set does not have any common data points. We find the OLS estimator $\hat{\boldsymbol{\beta}}_\lambda^k$ based on the training data $(\boldsymbol{Z}, \boldsymbol{W})$ and support $\hat{S}_\lambda$ (which was denoted by $\hat{\beta}_{S_j}^k$ for $\lambda = \lambda_j$ in the $\text{UoI}_{\text{Lasso}}^{\text{stable}}$ algorithm). For the $k^{th}$ iteration, the best $\lambda \in \boldsymbol{\lambda}$ is chosen by minimizing the prediction error for new validation data set $(\boldsymbol{Z}_0, \boldsymbol{W}_0)$,

$$\hat{\lambda}_k^{best} = \operatorname{argmin}_{\lambda \in \boldsymbol{\lambda}} ||\boldsymbol{Z}_0 - \boldsymbol{W}_0 \hat{\boldsymbol{\beta}}_\lambda^k||_2^2$$

So, the estimator for the $k^{th}$ iteration, $\hat{\beta}_S^k$ is redefined as $\hat{\beta}_{\hat{\lambda}_k^{best}}^k$. After $B_2$ number of bootstraps, the UoI selected variables set can be represented as

$$\hat{S}_{B_2}^{UoI} = \cup_{k=1}^{B_2} \hat{S}_{\hat{\lambda}_k^{best}} \tag{12}$$

where, $\hat{S}_{\hat{\lambda}_k^{best}}$ is the set of predictors selected at the Intersection step for the best penalization parameter $\hat{\lambda}_k^{best}$ for bootstrap iteration $k$ for Bagging.

For each resample, we have $\boldsymbol{W}$ as the the new design matrix. We normalize the columns so that, $\sum_{i=1}^m W_{ij}^2 = 1$ for $j = 1, \ldots, p$. In order to simplify the proof, we consider that the random variables $\boldsymbol{X}$ and $\boldsymbol{\varepsilon}$ have compact supports. Since, both $||\boldsymbol{X}||_2^2$ and $\boldsymbol{\varepsilon}^2$ are defined on separable Euclidean spaces, any probability measure on such space is tight. So, for any $\epsilon$, there exists a compact set $K_\epsilon$ such that $||\boldsymbol{X}||_2^2$ and $\boldsymbol{\varepsilon}^2$ has at least $1 - \epsilon$ probability on the compact support. So, without loss of generality, we can assume $||\boldsymbol{X}||_2^2$ and $\boldsymbol{\varepsilon}^2$ have compact supports and $||\boldsymbol{X}||_2^2$ and $\boldsymbol{\varepsilon}^2$ have finite moment generating functions (i.e., assumption (A1)).

The proof of the Theorem can be divided into three main lemma: Lemma 1, Lemma 2 and Lemma 3. The Lemma 1 deals with the model selection step. This Lemma is actually on the stability selection step. Lemma 1 follows directly from Theorem 2 presented in [33] on stability selection.

**Lemma 1.** *(Meinshausen and Bühlmann (2010) [33]) Consider the model in* (3). *Consider $\alpha$ given by $\alpha^2 = \nu c_{min}(m)/m$, for any $\nu \in ((7/\kappa)^2, 1/\sqrt{2})$, and $m = Cs^2$. Let $a_n$ be a sequence with $a_n \to \infty$*

*for* $n \to \infty$. *Let* $\lambda_{min} = 2\sigma(\sqrt{2C}s + 1)\sqrt{\log(p \wedge a_n)/n}$. *Assume that* $p > 10$ *and* $s \geq 7$ *and that the assumptions (A1)-(A3) is satisfied. Then there exists some* $\delta = \delta_s \in (0, 1)$ *such that for all* $\pi_{thr} \geq 1 - \delta$ *and data* $(\boldsymbol{Y}_i, \boldsymbol{X}_i)_{i=1}^n$ *belonging to the set* $\Omega_A$ *with* $P(\Omega_A) \geq p_s(1 - 5/(p \wedge a_n))$, *no noise variables are selected,*

$$N \cap \hat{S}_\lambda^{stable} = \emptyset,$$

*where* $\hat{S}_\lambda^{stable} =$ *support selected by stability selection with* $\lambda \geq \lambda_{min}$. *For data* $(\boldsymbol{Y}_i, \boldsymbol{X}_i)_{i=1}^n$ *belonging to the same set* $\Omega_A$,

$$(S \backslash S_{small;\lambda}) \subseteq \hat{S}_\lambda^{stable}$$

*where* $S_{small;\lambda} = \{k : |\beta_k| \leq 0.3(Cs)^{3/2}\lambda\}$. *This implies that all variables with sufficiently large regression coefficient are selected.*

The proof of Lemma 1 is in Meinshausen and Bühlmann (2010) [33]. The proof of part (a) of Theorem 1 follows directly from Lemma 1.

The Lemma 2 deals with the model estimation step. This lemma uses the results from Lemma 1 to give a bound on the false positives and false negatives after the model estimation step.

**Lemma 2.** *Consider the model in (3) and assumptions (A1)-(A4). Let* $a_n$ *be a sequence with* $a_n \to \infty$ *for* $n \to \infty$. *Let* $\lambda_{min} = 2\sigma(\sqrt{2C}s + 1)\sqrt{\log(p \wedge a_n)/n}$, *for some constant* $C > 1$. *For a set* $\Omega$ *such that* $P(\Omega) \geq \min\{1 - (1 - p_W)^{B_2}, (p_W)^{B_2}\}$, *where,* $p_W = p_s^2(1 - 5/(p \wedge a_n))$ *and data* $(\boldsymbol{Y}_i, \boldsymbol{X}_i)_{i=1}^n$ *as element of the set* $\Omega$, *we have,*

$$N \cap \hat{S}_{B_2}^{UoI} = \emptyset$$

*and*

$$(S \backslash S_{small;\lambda_{min}}) \subseteq \hat{S}_{B_2}^{UoI}$$

*where,* $S_{small;\lambda_{min}} = \{k : |\beta_k| \leq 0.3(Cs)^{3/2}\lambda_{min}\}$.

The proof of Lemma 2 is given in Section A.4.3. The proof of Lemma 2 goes in two steps, first the properties of the best selected regularization parameter $\hat{\lambda}^{best}$ is provided (Lemma 4) and second the bounds of Type I error (False Positives) and Type II error (False Negatives) are provided for the ordinary least squares (OLS) coefficient estimate, $\hat{\boldsymbol{\beta}}_{\hat{\lambda}^{best}}$ based on the best regularization parameter $\hat{\lambda}^{best}$.

The Lemma 3 deals with the estimation accuracy of the regression coefficient estimates obtained from the UoI$_{Lasso}^{stable}$ algorithm. This Lemma proves part (c) of Theorem 1. The proof of this Lemma also depends on Lemma 2.

**Lemma 3.** *Consider the model in (3) and assumptions (A1)-(A4). Let* $a_n$ *be a sequence with* $a_n \to \infty$ *for* $n \to \infty$. *Let* $\lambda_{min} = 2\sigma(\sqrt{2C}s + 1)\sqrt{\log(p \wedge a_n)/n}$, *for some constant* $C > 1$. *For a*

*set $\Omega$ such that $\mathrm{P}(\Omega) \geq \min\left\{1 - (1 - p_W)^{B_2}, (p_W)^{B_2}\right\}$, where, $p_W = p_s^2(1 - 5/(p \wedge a_n))$ and data $(\boldsymbol{Y}_i, \boldsymbol{X}_i)_{i=1}^n$ as element of the set $\Omega$, we have,*

$$\left|\left|X(\hat{\beta}_{UoI}^{stable} - \beta)\right|\right|^2 \leq C_1 \sigma^2 \frac{\log p}{n} \tag{13}$$

*for some constant $C_1 > 0$.*

The proof of Lemma 3 is given in Section A.4.4. The proof of Lemma 3 makes use of Lemma 2.

### A.4.3    Proof of Lemma 2

The proof of Lemma 2 uses Lemma 4. Let $\hat{\boldsymbol{\beta}}_{UoI}^{stable}$ be the UoI-estimate of the coefficient parameter $\boldsymbol{\beta}$ after the intersection step using stability selection and union step as in the UoI algorithm with $B_2$ bootstrap samples.

From the Lemma 4, we get that, for $\lambda > \lambda_{min}$,

$$\mathbb{E}\left(||\boldsymbol{W}_{0\lambda}\hat{\boldsymbol{\beta}}_\lambda - \boldsymbol{W}_{0\lambda}\boldsymbol{\beta}_\lambda||_2^2\right) \leq (q_\lambda + c_1/\sqrt{n}) + 0.3C^3(c_2 + c_3/\sqrt{n})ns^6\lambda^2$$

with probability $p_W := p_s^2(1 - 5/(p \wedge a_n))$ for some constants $C, c_1, c_2$ and $c_3$. Now, for $\lambda > \lambda_{min}$, $q_\lambda \leq s$. So, the upper bound is minimized for $\hat{\lambda}^{best} = C\lambda_{min}$.

Now, we already have that, with high probability (from Lemma 1), for $\lambda > \lambda_{min}$,

$$(S \backslash S_{small;\lambda}) \subseteq \hat{S}_\lambda^{stable}$$

where $S_{small;\lambda} = \{k : |\beta_k| \leq 0.3(Cs)^{3/2}\lambda\}$. Now, repeating the resampling procedure of Union step for $B_2$ times, even if the selected $\hat{\lambda}^{best} = O(\lambda_{min})$ for one case, we have the support corresponding to $\hat{S}_{\lambda_{min}}^{stable}$.

So, we have with probability, $(1 - (1 - p_W)^{B_2})$,

$$(S \backslash S_{small;\lambda_{min}}) \subseteq \hat{S}_{B_2}^{UoI}$$

where $S_{small;\lambda_{min}} = \{k : |\beta_k| \leq 0.3(Cs)^{3/2}\lambda_{min}\}$.

Also, from the Lemma 1, we get that for selected variables from stability selection for $\lambda > 0$, $\hat{S}_\lambda^{stable}$, we have with high probability,

$$N \cap \hat{S}_\lambda^{stable} = \emptyset,$$

$\lambda \geq \lambda_{min}$. Now, after the Union step with $B_2$ resamples, we have the same property, if $\hat{\lambda}^{best} \geq \lambda_{min}$ for each of the iterations. For each iteration, the event happens with probability $p_W = p_s^2(1 - 5/(p \wedge a_n))$. So, we have that with probability, $p_W^{B_2}$,

$$N \cap \hat{S}_{B_2}^{UoI} = \emptyset.$$

Lastly, we go into Lemma 4.

**Lemma 4.** *Consider the model in (3) and assumptions (A1)-(A4). Let $a_n$ be a sequence with $a_n \to \infty$ for $n \to \infty$. Let $\lambda_{min} = 2\sigma(\sqrt{2C}s+1)\sqrt{\log(p \wedge a_n)/n}$, for some constant $C > 1$. Consider the bootstrap resample $(\boldsymbol{Z}, \boldsymbol{W})$ from $(\boldsymbol{Y}, \boldsymbol{X})$ and the estimated coefficient $\hat{\boldsymbol{\beta}}_\lambda$ (for any $\lambda \in \boldsymbol{\lambda}$, where, $\boldsymbol{\lambda}$ is a set of regulization parameters) from model selection step, under the assumptions of Theorem 1. Then, the best $\lambda$, denoted by $\hat{\lambda}^{best}$ selected by minimizing prediction error, $||\boldsymbol{Z}_0 - \boldsymbol{W}_0\hat{\boldsymbol{\beta}}_\lambda||_2$, for the validation data set $(\boldsymbol{Z}_0, \boldsymbol{W}_0)$ has the property that $\hat{\lambda}^{best} \geq \lambda_{min}$ with probability greater than or equal to $p_s^2(1 - 5/(p \wedge a_n))$.*

*Proof.* We have $(\boldsymbol{Z}, \boldsymbol{W})$ as the bootstrap training sample from $(\boldsymbol{Y}, \boldsymbol{X})$ and $(\boldsymbol{Z}_0, \boldsymbol{W}_0)$ as the bootstrap validation sample from $(\boldsymbol{Y}, \boldsymbol{X})$. Let us define, $\hat{S}_\lambda := \{j : \hat{\boldsymbol{\beta}}_\lambda \neq 0\}$ for any $\lambda \in \boldsymbol{\lambda}$. Let, $\hat{\boldsymbol{\beta}}_\lambda^0$ be $\hat{\boldsymbol{\beta}}_\lambda$ restricted to $\hat{S}_\lambda$, $\boldsymbol{\beta}_\lambda$ be $\boldsymbol{\beta}$ restricted to $\hat{S}_\lambda$ and $\boldsymbol{\beta}_r$ is the set $\{\beta_j | j \notin \hat{S}_\lambda\}$. Also, $\boldsymbol{W}_\lambda$ is $\boldsymbol{W}$ restricted to the columns corresponding to $\hat{S}_\lambda$, $\boldsymbol{W}_{0\lambda}$ is $\boldsymbol{W}_0$ restricted to the columns corresponding to $\hat{S}_\lambda$ and $\boldsymbol{W}_r$ is $\boldsymbol{W}$ restricted to the columns corresponding to $\mathcal{S}\setminus\hat{S}_\lambda$. So, $\hat{\boldsymbol{\beta}}_\lambda^0 = (\boldsymbol{W}_\lambda^T\boldsymbol{W}_\lambda)^{-1}\boldsymbol{W}_\lambda^T\boldsymbol{Z}$ and $s^\lambda := |\hat{S}^\lambda \cap S|$, $n^\lambda := |\hat{S}^\lambda \cap S^C|$ and $q^\lambda := s^\lambda + n^\lambda$. Recall that, $S = \{j | \beta_j \neq 0\}$. At first, we shall consider, $\lambda \geq \lambda_{min}$, such that, $q_\lambda < n$ and $q_\lambda < s$ from the model selection step.

The prediction error for validation data set $(\boldsymbol{Z}_0, \boldsymbol{W}_0)$ is $||\boldsymbol{W}_0\hat{\boldsymbol{\beta}}_\lambda^0 - \boldsymbol{W}_0\beta_\lambda||_2^2$. Now, conditional on $\boldsymbol{W}$,

$$
\begin{aligned}
\hat{\boldsymbol{\beta}}_\lambda^0 &= (\boldsymbol{W}_\lambda^T\boldsymbol{W}_\lambda)^{-1}\boldsymbol{W}_\lambda^T\boldsymbol{Z} \\
&= (\boldsymbol{W}_\lambda^T\boldsymbol{W}_\lambda)^{-1}\boldsymbol{W}_\lambda^T(\boldsymbol{W}\boldsymbol{\beta}_\lambda + \boldsymbol{\varepsilon}) \\
&= (\boldsymbol{W}_\lambda^T\boldsymbol{W}_\lambda)^{-1}\boldsymbol{W}_\lambda^T(\boldsymbol{W}_\lambda\boldsymbol{\beta}_\lambda + \boldsymbol{W}_r\boldsymbol{\beta}_r + \boldsymbol{\varepsilon}) \\
&= \boldsymbol{\beta}_\lambda + (\boldsymbol{W}_\lambda^T\boldsymbol{W}_\lambda)^{-1}\boldsymbol{W}_\lambda^T\boldsymbol{W}_r\boldsymbol{\beta}_r + (\boldsymbol{W}_\lambda^T\boldsymbol{W}_\lambda)^{-1}\boldsymbol{W}_\lambda^T\boldsymbol{\varepsilon}.
\end{aligned}
$$

So,
$$
\boldsymbol{W}_{0\lambda}\hat{\boldsymbol{\beta}}_\lambda^0 = \boldsymbol{W}_{0\lambda}\boldsymbol{\beta}_\lambda + \boldsymbol{W}_{0\lambda}(\boldsymbol{W}_\lambda^T\boldsymbol{W}_\lambda)^{-1}\boldsymbol{W}_\lambda^T\boldsymbol{W}_r\beta_r + \boldsymbol{W}_{0\lambda}(\boldsymbol{W}_\lambda^T\boldsymbol{W}_\lambda)^{-1}\boldsymbol{W}_\lambda^T\boldsymbol{\varepsilon}
$$

So,
$$
\begin{aligned}
\mathbb{E}\left(\boldsymbol{W}_{0\lambda}\hat{\boldsymbol{\beta}}_\lambda^0 - \boldsymbol{W}_{0\lambda}\boldsymbol{\beta}_\lambda \Big| \boldsymbol{W}\right) &= \boldsymbol{W}_{0\lambda}(\boldsymbol{W}_\lambda^T\boldsymbol{W}_\lambda)^{-1}\boldsymbol{W}_\lambda^T\boldsymbol{W}_r\boldsymbol{\beta}_r = \mu_0 \ \ (\text{say}) \\
\mathrm{Var}\left(\boldsymbol{W}_{0\lambda}\hat{\boldsymbol{\beta}}_\lambda^0 - \boldsymbol{W}_{0\lambda}\boldsymbol{\beta}_\lambda \Big| \boldsymbol{W}\right) &= \sigma^2\boldsymbol{W}_{0\lambda}(\boldsymbol{W}_\lambda^T\boldsymbol{W}_\lambda)^{-1}\boldsymbol{W}_\lambda^T\boldsymbol{W}_\lambda(\boldsymbol{W}_\lambda^T\boldsymbol{W}_\lambda)^{-1}\boldsymbol{W}_{0\lambda}^T \\
&= \sigma^2\boldsymbol{W}_{0\lambda}(\boldsymbol{W}_\lambda^T\boldsymbol{W}_\lambda)^{-1}\boldsymbol{W}_{0\lambda}^T
\end{aligned}
$$

So, using results on expectation of quadratic forms of Gaussian random variables,

$$
\mathbb{E}\left(||\boldsymbol{W}_{0\lambda}\hat{\boldsymbol{\beta}}_\lambda^0 - \boldsymbol{W}_{0\lambda}\boldsymbol{\beta}_\lambda||_2^2 \Big| \boldsymbol{W}\right) = tr\left(\boldsymbol{W}_{0\lambda}(\boldsymbol{W}_\lambda^T\boldsymbol{W}_\lambda)^{-1}\boldsymbol{W}_{0\lambda}^T\right)\sigma^2 + \mu_0^T\mu_0
$$

Now, by matrix Hoeffding inequality on the covariance matrix as shown in [1], $tr\left((\boldsymbol{W}_\lambda^T\boldsymbol{W}_\lambda)^{-1}\boldsymbol{W}_{0\lambda}^T\boldsymbol{W}_{0\lambda}\right)$ can be replaced by $tr(\Sigma_\lambda^{-1}\Sigma_\lambda) = q_\lambda$ at cost $O(n^{-1/2})$ with high probability, that is,

$$
\mathrm{P}\left(\sqrt{n}\left|tr\left((\boldsymbol{W}_\lambda^T\boldsymbol{W}_\lambda)^{-1}\boldsymbol{W}_{0\lambda}^T\boldsymbol{W}_{0\lambda}\right) - q_\lambda\right| < c\right) \geq 1 - k_1\exp(-k_2c^2)
$$

for constants $k_1$ and $k_2$. So, for some constant $c_1 > 0$, with probability at least $(1 - k_1 \exp(-k_2 c^2))$,

$$tr\left(\boldsymbol{W}_{0\lambda}(\boldsymbol{W}_\lambda^T \boldsymbol{W}_\lambda)^- \boldsymbol{W}_{0\lambda}^T\right) \leq q_\lambda + c/\sqrt{n}$$

Now, for the second term,

$$
\begin{aligned}
||\mu_0||_2^2 &= ||\boldsymbol{W}_{0\lambda}(\boldsymbol{W}_\lambda^T \boldsymbol{W}_\lambda)^{-1}\boldsymbol{W}_\lambda^T \boldsymbol{W}_r \boldsymbol{\beta}_r||_2^2 \\
&\leq \frac{\phi_{max}(\boldsymbol{W}_\lambda^T \boldsymbol{W}_\lambda)}{\phi_{min,+}^2(\boldsymbol{W}_\lambda^T \boldsymbol{W}_\lambda)} \phi_{max}(\boldsymbol{W}_{0\lambda}^T \boldsymbol{W}_{0\lambda})\phi_{max}(\boldsymbol{W}_r^T \boldsymbol{W}_r)||\boldsymbol{\beta}_r||_2 \\
&\leq \frac{\phi_{max}(\boldsymbol{W}_\lambda^T \boldsymbol{W}_\lambda)}{\phi_{min,+}^2(\boldsymbol{W}_\lambda^T \boldsymbol{W}_\lambda)} \phi_{max}(\boldsymbol{W}_{0\lambda}^T \boldsymbol{W}_{0\lambda})\phi_{max}(\boldsymbol{W}_r^T \boldsymbol{W}_r)||\boldsymbol{\beta}_r||_1
\end{aligned}
$$

Again, by matrix Hoeffding inequality on the covariance matrix as shown in [1] and the assumption (A3) on the covariance matrix of the explanatory variables, we have with probability at least $p_s(1 - k_3 \exp(-k_4 c^2))$ (for some constants $k_1$ and $k_2$),

$$\frac{\phi_{max}(\boldsymbol{W}_\lambda^T \boldsymbol{W}_\lambda)}{\phi_{min,+}^2(\boldsymbol{W}_\lambda^T \boldsymbol{W}_\lambda)} \phi_{max}(\boldsymbol{W}_{0\lambda}^T \boldsymbol{W}_{0\lambda}) \leq \frac{\phi_{max}^2(\Sigma_\lambda)}{\phi_{min}^2(\Sigma_\lambda)} + c/\sqrt{n}$$

$\phi_{max}(\boldsymbol{W}_r^T \boldsymbol{W}_r) \leq ns^2$ using the property $\sum_{i=1}^m W_{ij}^2 = 1$. So, for some constants $c_2, c_3 > 0$

$$||\mu_0||_2^2 \leq (c_2 + c_3/\sqrt{n})ns^2 ||\boldsymbol{\beta}_r||_1^2$$

and using Lemma 1,

$$\mathbb{E}\left(||\boldsymbol{W}_{0\lambda}\hat{\boldsymbol{\beta}}_\lambda - \boldsymbol{W}_{0\lambda}\boldsymbol{\beta}_\lambda||_2^2\right) \leq (q_\lambda + c_1/\sqrt{n}) + 0.3C^3(c_2 + c_3/\sqrt{n})ns^6 \lambda^2$$

We can observe that for $\lambda \geq \lambda_{min}$, $q_\lambda = s$ and $N \cap \hat{S}_\lambda = \emptyset$ and $(S\backslash S_{small;\lambda} \subseteq \hat{S}_\lambda)$ with high probability from Lemma 1. So, for constants $c_4$ and $c_5$, $\lambda \geq \lambda_{min}$ with probability greater than $p_s^2(1 - 5/(p \wedge a_n))$,

$$\mathbb{E}(||\boldsymbol{W}_{0\lambda}\hat{\boldsymbol{\beta}}_\lambda - \boldsymbol{W}_{0\lambda}\boldsymbol{\beta}_\lambda||_2^2) \leq (c_4 + c_5/\sqrt{n})ns^6 \lambda^2$$

For $\lambda < \lambda_{min}$, from [1], we get that, $q_\lambda = O(p)$. So, $\hat{\boldsymbol{\beta}}_\lambda^0 = (\boldsymbol{W}_\lambda^T \boldsymbol{W}_\lambda)^- \boldsymbol{W}_\lambda^T \boldsymbol{Z}$, where, $A^-$ is the generalized Moore-Penrose inverse of a matrix. Now, conditional on $\boldsymbol{W}$,

$$
\begin{aligned}
\hat{\boldsymbol{\beta}}_\lambda^0 &= (\boldsymbol{W}_\lambda^T \boldsymbol{W}_\lambda)^- \boldsymbol{W}_\lambda^T \boldsymbol{Z} \\
&= (\boldsymbol{W}_\lambda^T \boldsymbol{W}_\lambda)^- \boldsymbol{W}_\lambda^T (\boldsymbol{W}\boldsymbol{\beta}_\lambda + \boldsymbol{\varepsilon}) \\
&= (\boldsymbol{W}_\lambda^T \boldsymbol{W}_\lambda)^- \boldsymbol{W}_\lambda^T (\boldsymbol{W}_\lambda \boldsymbol{\beta}_\lambda + \boldsymbol{W}_r \boldsymbol{\beta}_r + \boldsymbol{\varepsilon}) \\
&= \boldsymbol{U}\boldsymbol{\beta}_\lambda + (\boldsymbol{W}_\lambda^T \boldsymbol{W}_\lambda)^- \boldsymbol{W}_\lambda^T \boldsymbol{W}_r \boldsymbol{\beta}_r + (\boldsymbol{W}_\lambda^T \boldsymbol{W}_\lambda)^- \boldsymbol{W}_\lambda^T \boldsymbol{\varepsilon}.
\end{aligned}
$$

where, $\boldsymbol{U} = \sum_{j:\mu_j \neq 0} \mu_j \boldsymbol{u}_j \boldsymbol{u}_j^T$, $\{\mu_j\}$ and $\{\boldsymbol{u}_j\}$ are eigenvalues and eigenvectors of $(\boldsymbol{X}^T \boldsymbol{X})$. So,

$$\boldsymbol{W}_{0\lambda}\hat{\boldsymbol{\beta}}_\lambda^0 = \boldsymbol{W}_{0\lambda}(\boldsymbol{U} - \boldsymbol{I})\boldsymbol{\beta}_\lambda + \boldsymbol{W}_{0\lambda}(\boldsymbol{W}_\lambda^T \boldsymbol{W}_\lambda)^{-1}\boldsymbol{W}_\lambda^T \boldsymbol{W}_r \boldsymbol{\beta}_r + \boldsymbol{W}_{0\lambda}(\boldsymbol{W}_\lambda^T \boldsymbol{W}_\lambda)^{-1}\boldsymbol{W}_\lambda^T \boldsymbol{\varepsilon}$$

So,

$$\mathbb{E}\left(\boldsymbol{W}_{0\lambda}\hat{\boldsymbol{\beta}}_{\lambda}^{0} - \boldsymbol{W}_{0\lambda}\boldsymbol{\beta}_{\lambda}\Big|\boldsymbol{W}\right) = \boldsymbol{W}_{0\lambda}(\boldsymbol{U}-\boldsymbol{I})\boldsymbol{\beta}_{\lambda} + \boldsymbol{W}_{0\lambda}(\boldsymbol{W}_{\lambda}^{T}\boldsymbol{W}_{\lambda})^{-1}\boldsymbol{W}_{\lambda}^{T}\boldsymbol{W}_{r}\boldsymbol{\beta}_{r} = \mu_1 + \mu_0 \;\; \text{(say)}$$

$$\mathrm{Var}\left(\boldsymbol{W}_{0\lambda}\hat{\boldsymbol{\beta}}_{\lambda}^{0} - \boldsymbol{W}_{0\lambda}\boldsymbol{\beta}_{\lambda}\Big|\boldsymbol{W}\right) = \sigma^2 \boldsymbol{W}_{0\lambda}(\boldsymbol{W}_{\lambda}^{T}\boldsymbol{W}_{\lambda})^{-1}\boldsymbol{W}_{\lambda}^{T}\boldsymbol{W}_{\lambda}(\boldsymbol{W}_{\lambda}^{T}\boldsymbol{W}_{\lambda})^{-1}\boldsymbol{W}_{0}^{T}$$

$$= \sigma^2 \boldsymbol{W}_{0\lambda}(\boldsymbol{W}_{\lambda}^{T}\boldsymbol{W}_{\lambda})^{-1}\boldsymbol{W}_{0\lambda}^{T}$$

So,

$$\mathbb{E}\left(||\boldsymbol{W}_{0\lambda}\hat{\boldsymbol{\beta}}_{\lambda}^{0} - \boldsymbol{W}_{0\lambda}\boldsymbol{\beta}_{\lambda}||_{2}^{2}\Big|\boldsymbol{W}\right) = tr\left(\boldsymbol{W}_{0\lambda}(\boldsymbol{W}_{\lambda}^{T}\boldsymbol{W}_{\lambda})^{-1}\boldsymbol{W}_{0\lambda}^{T}\right)\sigma^2 + (\mu_1+\mu_0)^{T}(\mu_1+\mu_0)$$

Now again, $tr\left((\boldsymbol{W}_{\lambda}^{T}\boldsymbol{W}_{\lambda})^{-1}\boldsymbol{W}_{0\lambda}^{T}\boldsymbol{W}_{0\lambda}\right)$ is of the order $O(n)$ with high probability and $(\mu_1+\mu_0)^{T}(\mu_1+\mu_0) \geq 0$. So, for some constant $C_3 > 0$,

$$\mathbb{E}(||\boldsymbol{W}_{0\lambda}\hat{\boldsymbol{\beta}}_{\lambda} - \boldsymbol{W}_{0\lambda}\boldsymbol{\beta}_{\lambda}||_{2}^{2}) \geq C_3 n$$

So, for $\lambda_1 < \lambda_{min}$ and $\lambda_2 \geq \lambda_{min}$, we have, with high probability, for large $n$,

$$\mathbb{E}(||\boldsymbol{W}_{0\lambda_1}\hat{\boldsymbol{\beta}}_{\lambda_1} - \boldsymbol{W}_{0\lambda_1}\boldsymbol{\beta}_{\lambda_1}||_{2}^{2}) \geq C_3 n > (c_4 + c_5/\sqrt{n})n s^6 \lambda_2^2 \geq \mathbb{E}(||\boldsymbol{W}_{0\lambda_2}\hat{\boldsymbol{\beta}}_{\lambda_2} - \boldsymbol{W}_{0\lambda_2}\boldsymbol{\beta}_{\lambda_2}||_{2}^{2})$$

So, for suitably chosen $c_4, c_5$ and $C_3$, the best selected $\lambda$ will have the property, $\lambda \geq \lambda_{min}$ with probability greater than $p_s^2(1 - 5/(p \wedge a_n))$. $\qquad\square$

### A.4.4    Proof of Lemma 3

The proof mostly follows from Lemma 2. Note that from Lemma 2, we get that,

$$N \cap \hat{S}_{B_2}^{UoI} = \emptyset \text{ and } (S \backslash S_{small;\lambda_{min}}) \subseteq \hat{S}_{B_2}^{UoI}$$

where, $S_{small;\lambda_{min}} = \{k : |\beta_k| \leq 0.3(Cs)^{3/2}\lambda_{min}\}$, for some constant $C > 1$ with probability $\min\left\{1 - (1 - p_W)^{B_2}, (p_W)^{B_2}\right\}$. So, for $\mathcal{S} = \{1, \ldots, p\}$ and

$$\begin{aligned}
||\hat{\beta}_{UoI}^{stable} - \beta||_1 &= ||(\hat{\beta}_{UoI}^{stable})_{\hat{S}_{B_2}^{UoI}} - \beta_{\hat{S}_{B_2}^{UoI}}||_1 + ||(\hat{\beta}_{UoI}^{stable})_{\mathcal{S}\backslash\hat{S}_{B_2}^{UoI}} - \beta_{\mathcal{S}\backslash\hat{S}_{B_2}^{UoI}}||_1 \\
&= ||(\hat{\beta}_{UoI}^{stable})_{\hat{S}_{B_2}^{UoI}} - \beta_{\hat{S}_{B_2}^{UoI}}||_1 + ||(\hat{\beta}_{UoI}^{stable})_{\mathcal{S}\backslash S_{B_2}^{UoI}} - \beta_{S_{small;\lambda_{min}}\backslash\hat{S}_{B_2}^{UoI}}||_1 \\
&\leq c_1\sigma\frac{s}{\sqrt{n}} + c_2\sigma\sqrt{\frac{s^7\log p}{n}}
\end{aligned}$$

for some constants $c_1, c_2 > 0$. So, by norm inequality, we get that,

$$||\hat{\beta}_{UoI}^{stable} - \beta||_2^2 \leq c_3\sigma^2 s^7 \frac{\log p}{n}$$

From which, if we are considering $s$ is constant, by matrix norm inequalities and bound on $||X||_2^2$ due to compact support, we can state that,

$$||X(\hat{\beta}_{UoI}^{stable} - \beta)||_2^2 \leq C_1\sigma^2\frac{\log p}{n}$$

for some constant $C_1 > 0$, where, $C_1$ depend on $s$, the constant $C$ and the cumulant generating function of $||X||_2^2$.

### A.4.5 BoLasso Algorithm

In the section A.2, we propose the $UoI_{Lasso}$ algorithm. In the $UoI_{Lasso}$ algorithm, two main parts are model selection and model estimation. The model selection in the $UoI_{Lasso}$ algorithm is performed using BoLasso algorithm [1]. Bach (2008) [1] provide the result on model consistency of the support selected by the BoLasso algorithm. The assumptions considered in [1] were

(B1) The cumulant generating functions $\mathbb{E}(\exp(s||X||_2^2)$ and $\mathbb{E}(\exp(sY^2)$ are finite for some $s > 0$.

(B2) The joint matrix of second order moments $Q = \mathbb{E}(XX^T) \in \mathbb{R}^{p \times p}$ is invertible.

(B3) $\mathbb{E}(Y|X) = X^T \boldsymbol{w}$ and $\mathrm{Var}(Y|X) = \sigma^2$ a.s. for some $\boldsymbol{w} \in \mathbb{R}^p$ and $\sigma \in \mathbb{R}_+$, where, $\mathbb{R}_+ = \{x \in \mathbb{R}|x > 0\}$.

Let us consider that for regularization parameter, $\lambda \in \mathbb{R}_+$, support recovered by BoLasso, that is the number of non-zero regression coefficients for BoLasso, is $\hat{S}_\lambda^{BoLasso}$. Like in section A.2, the BoLasso support for $\lambda_j$ is $S_j \equiv \hat{S}_{\lambda_j}^{BoLasso}$. If we consider the model selection step of $UoI_{Lasso}$ algorithm with $B_1$ bootstrap resamples, Bach (2008) [1] demonstrated the dependence of probability of correct model selection on $B_1$. The result as established in [1] is

**Lemma 6.** *(Bach (2008)) Consider model in* (3) *and assumptions (B1)-(B3). Then, for $\lambda_n = Cn^{-1/2}$, $C > 0$, the probability that the BoLasso does not exactly select the correct model, i.e., for all $B_1 > 0$, $\mathrm{P}(\hat{S}_\lambda^{BoLasso} \neq S)$ has the following upper bound:*

$$\mathrm{P}(\hat{S}_\lambda^{BoLasso}) \leq B_1 A_1 e^{-A_2 n} + A_3 \frac{\log n}{n^{1/2}} + A_4 \frac{\log B_1}{B_1}, \tag{14}$$

*where $A_1, A_2, A_3, A_4$ are strictly positive constants.*

As, we can see from the Lemma 6, that for correct model selection with high probability, we should have $B_1 \to \infty$ but at a rate slower than $\log n$ for BoLasso.

However, for theoretical analysis of the model estimation step of $UoI_{Lasso}$ algorithm, we need separate control on the false positives and false negatives of the recovered support. In Lemma 6, the result gives a bound on total error in support recovery, but not separate control on false positives and false negatives. For this reason, in the theoretical analysis of UoI methods, in stead of using BoLasso in the model selection step, we use stability selection [33].

Note that, we conjecture that in BoLasso, increase in $B_1$ increases the probability of no false positives but increase of $B_1$ at fast rate also decreases the probability of false negative selection. One of our future works would be to explicitly show such behavior exists for BoLasso and then use those results to theoretically analyze $UoI_{Lasso}$ method starting with BoLasso in stead of stability selection method.

## A.5 $UoI_{Lasso}$ Outperforms Other Methods: Expanded Results for Simulated Data

Here, we extend the results from the example simulation that were presented in Fig. 2 in the main text. To remind the reader, we compared $UoI_{Lasso}$ (black) to five other model estimation methods: Ridge (purple), Lasso (green), SCAD (red), BoATS (blue), and debiased Lasso (pink). We quantified several metrics of both model recovery (i.e., selection accuracy and estimation error) and prediction quality (accuracy: $R^2$; parsimony: Bayesian Information Criterion). The expanded results presented in Fig. 6 are for a simulated data set generated from a model with parameters distributed as the grey histogram in Fig. 6(b). In particular, there were $n = 1200$ examples with observation variables ($y$) generated from Eqn. (1), with $p = 300$ total parameters ($n/p = 4$), $k = 100$ non-zero parameters (sparsity: $1 - k/p = 0.66$) that were symmetrically distributed with exponentially increasing frequency as a function of parameter magnitude, and noise magnitude of $\sigma^2 = 0.2 \times \sum_j |\beta_j|$. We took statistics of the metrics across 100 randomized cross-validation samples of the data.

Fig. 6(a) shows scatter plots of predicted vs. actual values of the observation variable on held-out data samples. Fig. 6(b) displays histograms of final model parameters (colors) overlaid on actual model parameters (grey). Fig. 6(c) plots the mean $\pm$ s.d. of the estimate (colors) for each model parameter, ordered by the actual value of that parameter (actual values shown in grey). Fig. 6(d) shows the variability of the estimated values as a function of their magnitude. Fig. 6(e) quantifies a variety of properties of the results of each algorithm. Below, we summarize the results for each method, and we provide some intuition as to why these results are observed. This provides insight into the general superiority of the $UoI_{Lasso}$ algorithm.

Ridge regression (purple) gave very weak model selection (i.e., few parameters equal to 0) and parameter estimates that were highly biased towards smaller values. This resulted in poor selection, estimation, and prediction accuracy, and the worst prediction parsimony (i.e., prediction accuracy relative to number of non-zero model parameters). This is to be expected, as the $L_2$ norm used by the algorithm is not a sparsity-inducing regularizer. Because the actual model in this case is quite sparse (only one-in-three parameters are non-zero), the relatively large value of the regularization parameter "shrinks" the values of all parameters towards zero, resulting in large bias.

The least angle shrinkage and selection operator (Lasso, green) is the industry standard method for regularized estimation of parameters in sparse models. It gave results that were much better than ridge regression for many metrics. However, it did only modestly well compared to the other algorithms tested here. This is to be expected, as the $L_1$ norm used by Lasso is a sparsity inducing regularizer (i.e., unlike ridge, Lasso "shrinks" the values of all parameters towards zero to induce sparsity), but it imposes a Laplacian prior over the distribution of parameters (which in this example is known to be far from correct). That is, because the actual model in this example is both highly sparse and has many parameters with large magnitudes (counter to the Laplacian prior), Lasso

Figure 6: **Expanded range of observed results, in comparison with existing algorithms.** $UoI_{Lasso}$ outperforms other methods.

resulted in only modest estimation error and prediction accuracy.

The Smoothly Clipped Absolute Deviation estimator (SCAD, red) is widely considered to be the state-of-the-art for regularized linear regression: for a given regularization strength, the magnitude of estimation shrinkage is larger for small values than for large values. This should result in both model selection and reduced bias in the estimation of large parameters. We found that SCAD had good data prediction accuracy, but intermediary selection accuracy and estimation error, and very low variability. It is worth noting here that SCAD has two main computational disadvantages compared to the rest of the algorithms presented here: it has a two-dimensional hyper-parameter space (though in practice, one of them is held constant), and (more seriously) it requires solving a non-convex optimization problem, making stability of solutions and scaling to large data sets less straightforward.

The Bootstrapped Adaptive Threshold Selection (BoATS) algorithm (blue) takes a very simple approach to model selection and estimation. First, it gets an initial estimate of all parameter values; then it sets all parameters below a threshold to zero and re-estimates the remaining parameters with bagged OLS; and then it optimizes the parameter threshold to maximize prediction accuracy (Fig. 6(a)). This has the attractive properties of setting many parameters exactly to zero so as to optimize prediction accuracy, and it gives nearly unbiased and accurate estimates of the remaining values (Fig. 6(c)). However, because of the hard thresholding combined with process noise, its estimates around the threshold are highly variable (Fig. 6(d)).

In a somewhat similar approach to BoATS, a recently proposed method to de-bias Lasso estimates and then use statistical tests (i.e., thresholds based on $p$-values) for model selection (debiased Lasso, pink) was overly aggressive, setting more values to zero than should be. This resulted in the worst data prediction accuracy, although it achieved good selection accuracy, estimation error and prediction parsimony. This can be understood because the selection of unbiased parameter estimates according to an *a priori* arbitrary statistical criterion is done outside the context of optimizing prediction accuracy, and it sets many parameters exactly to zero.

$UoI_{Lasso}$ is designed to maximize prediction accuracy (Fig. 6(a), black) by first selecting the correct variables (Fig. 6(b), black), and then estimating their values with high accuracy (Fig. 6(c), black) and low variance (Fig. 6(d), black) with bagged OLS, which is nearly unbiased. It therefore offers the benefits of the strong selection algorithms (BoATS and debiased Lasso), but with the low variability of the structured regularizers (Lasso, SCAD), while simultaneously having accurate and nearly unbiased estimates. It also involves only calculations for convex optimizations, and so it scales very well.

To summarize these results, across all the algorithms we examined, we found that $UoI_{Lasso}$ (black) generally resulted in the highest selection accuracy (Fig. 6(e), right), with parameter estimates with lowest error (Fig. 6(e), right-center) and competitive variance (Fig. 6(e), center-right). In addition, it led to the best prediction accuracy (Fig. 6(e), center-left), with a small number of model parameters (Fig. 6(e), left-center), giving best prediction parsimony (Fig. 6(e), left).

## A.6  $UoI_{Lasso}$ Outperforms Other Methods: Simulated Data with Different Parameter Distributions and Sparsity Levels

We further examined the generality of the superior performance of $UoI_{Lasso}$ compared to other algorithms by simulating data generated by models with different underlying distributions. We varied both the distribution of the non-zero parameters (Fig. 7(a), black histograms) and the over-all sparsity of the model (Fig. 7(a), grey bar at zero). We kept constant: the number of non-zero parameters ($k = 100$ for all); the noise ($\sigma^2 = 0.2 \times \sum_j |\beta_j|$); and the number of samples ($n$) relative to the total number of model parameters ($p$) ($n/p = 3$). We simulated data generated from four different

Figure 7: **Simulations across different parameter distributions and levels of sparsity.** $UoI_{Lasso}$ outperforms other known methods. See the text for details and discussion.

distributions: exponentially decaying as a function of magnitude: roughly Laplacian (left); uniform (center left); exponentially increasing as a function of magnitude (center right); and clustered-positive (right). We varied the sparsity from 0 ($p = 100$ parameters total) to 0.9 ($p = 1000$ parameters total). Note that the number of samples in these simulations was different than for the example presented in Fig 2 and Fig. 6.

Generally speaking, across all distributions and levels of sparsity, $UoI_{Lasso}$ generally had lowest estimation error (Fig. 7(b), black), highest prediction accuracy (Fig. 7(c), black), and lowest estimation variability (Fig. 7(e), black). The hard-thresholding procedures (BoATS and debiased Lasso) generally had the highest variability (Fig. 7(e), pink and blue). Across distributions, ridge regression (purple) and debiased Lasso (pink) had the strongest dependencies on sparsity. However, for all distributions, all methods except for $UoI_{Lasso}$ exhibited systematic dependencies of selection accuracy on the degree of sparsity (Fig. 7(d)), while $UoI_{Lasso}$ was nearly independent. Thus, for fixed $B_1$ and $B_2$, the selection properties of $UoI_{Lasso}$ depend only on the shape of the underlying non-zero parameter distribution and the amount of noise in the process, and not the overall degree of sparsity. This was a unique property of $UoI_{Lasso}$. The superior performance of $UoI_{Lasso}$ on model estimation error and prediction accuracy despite reduced selection accuracies at low sparsities is due to the fact that the parameters that are getting set to zero are those that have very low magnitude and cannot be reliably estimated, given the amount of noise in the generating process.

## A.7 $UoI_{Lasso}$ Outperforms Other Methods: Simulated Data with Different Noise Magnitudes

To determine the robustness of $UoI_{Lasso}$ to the magnitude of process noise, we examined the performance of the different methods as the magnitude of the process noise increased. Specifically, we varied the standard deviation of the additive Gaussian noise as a multiple of the summed weight magnitude ($\sigma^2 = m \times \sum_j |\beta_j|$). The plots of Fig. 8(a)-(d) show results for data generated from the clustered model distribution (e.g., Fig. 7(a) right, sparsity: $1 - k/p = 0.66$), for six values of the multiplicative factor $m \in ([0 : 0.6])$.

As expected, all algorithms performed very well when there was no noise, and most generally performed worse with increasing noise levels: estimation error and variability increased, prediction accuracy and support overlap decreased. Importantly, though, $UoI_{Lasso}$ (black) generally performed as well as or better than the other algorithms, as the noise magnitude increased.

## A.8 $UoI_{L1Logistic}$ for Classification: Identifying Fewer Features without Loss of Prediction Accuracy

We have primarily demonstrated the power of the UoI method in the context of linear regression with the $UoI_{Lasso}$ algorithm. However, the base UoI framework is much more general, and it

Figure 8: **Simulations across different noise magnitudes.** $UoI_{Lasso}$ outperforms all other known methods. See the text for details and discussion.

can be applied to other regression problems, as well as other machine learning problems such as classification. To demonstrate this, we implemented a classification algorithm using logistic regression ($UoI_{L1Logistic}$), and we compared it to $L_1$-Logistic regression on three diverse biomedical data sets from the UCI data repository. In the Dorothea data set, the goal is to find a small number of features that are predictive of whether a pharmaceutical compound is active (binds to target receptor) or inactive (is non-binding); in the Arcene data set, the problem of feature detection for prediction of cancer is presented, where mass spectrometry data indicating protein levels is used to separate healthy individual from those with cancer; and in the Parkinson's Disease data set, the goal is to predict the stage of disease progression (a numerical score assigned by a clinician) from audio recordings of the patients speech. See Tables 1, 2, and 3 for a summary of our results. On all three of these data sets, in agreement with the results presented above on genetics, neuroscience, and synthetic data, we found that $UoI_{L1Logistic}$ performed well with respect to both prediction and parsimony. In particular, it gave equivalent or better prediction accuracy, with many fewer parameters (3, 5, and 10 respectively), resulting in the best prediction parsimony.

Table 1:

| Dorothea | Prediction Accuracy | Selection Ratio (PSR) | Parsimony (BIC) |
|---|---|---|---|
| $L_1$-Logistic | 93% | $53 \times 10^{-5}$ | 456 |
| $UoI_{L1Logistic}$ | 93% | $3 \times 10^{-5}$ | 174 |

Table 2:

| Arcene | Prediction Accuracy | Selection Ratio (PSR) | Parsimony (BIC) |
|---|---|---|---|
| $L_1$-Logistic | 66% | $59 \times 10^{-4}$ | 437 |
| $UoI_{L1Logistic}$ | 66% | $5 \times 10^{-4}$ | 280 |

Table 3:

| Parkinson's | Prediction Accuracy | Selection Ratio (PSR) | Parsimony (BIC) |
|---|---|---|---|
| $L_1$-Logistic | 65% | 0.69 | 533 |
| $UoI_{L1Logistic}$ | 68% | 0.31 | 478 |

## A.9 $UoI_{CUR}$ for Matrix Decomposition

One of the popular dimensionality reduction methods used in many applications is the column subset selection problem (CSSP) [9], a variant of which is the so-called CUR matrix decomposition [17, 30]. Given a large data matrix $A \in \mathbb{R}^{m \times n}$, whose columns we wish to select, suppose $V_k$ is the matrix consisting of the top $k$ right singular vectors of $A$. Then, the leverage score of the $i$th column of $A$ is given by

$$\ell_i = \frac{1}{k} \|V_k(i,:)\|_2^2, \tag{15}$$

i.e., by the norm of the $i$th row of $V_k$. In leverage score sampling, the columns of $A$ are sampled using the probability distribution $p_i = \min\{1, \ell_i\}$, where $\ell_i$ is given by Eqn. (15). Many popular methods for CSSP/CUR involve the use of this leverage score distribution as the importance sampling distribution with respect to which to sample columns [17, 9, 30]. (Importantly, while a naïve version of this algorithm is expensive, due to the computation of the SVD, the leverage scores of $A$ can be well-approximated in the time it takes to perform a random projection on the matrix $A$ [29, 16], and the leverage score method has been applied to very large data sets [29, 23].) In this work, we show how the UoI framework can be adapted to the CSSP/CUR matrix decomposition problem.

The basic $UoI_{CUR}$ *algorithm* is as follows. We consider the bootstrap resampling approach. We compute the different subsets of columns (and rows) $C_i$ for the different bootstrap samples $i = 1, \ldots, B_1$, and for different ranks $k$ using leverage score sampling.

- **Intersection Step:** We then intersect the support (indices) of the subsets of columns (and rows) $C_i$ over the bootstraps to obtain a smaller intersected subset $\hat{C}^{(k)}$ (for different ranks $k$). This intersection operation reduces the variance in sampling.

- **Union Step:** We then obtain a larger union set of columns by taking union of the intersected subsets $\hat{C}^{(k)}$ over different ranks $k$.

As an illustration of the $UoI_{CUR}$ algorithm, we have applied it to the analysis of genetics data. Analysis of gene expression DNA microarray data has become popular for studying a variety of biological processes [35]. In the microarray data, we have $m$ genes (from $m$ individuals, possibly from different populations) and a series of $n$ arrays probe the genome-wide expression levels in $n$ different samples, possibly under $n$ different experimental conditions. Hence, the data from microarray experiments can be naturally represented as a matrix $A \in \mathbb{R}^{m \times n}$, where $A_{ij}$ indicates whether the $j$th expression level exists for the $i$th gene. Typically, the matrix could have entries $\{-1, 0, 1\}$, indicating whether the expression exists ($\pm 1$) or not (0), with the sign indicating the order of the sequence.

Article [35] used the CUR decomposition with a greedy column selection algorithm to select a subset of gene expressions or single nucleotide polymorphisms (SNPs) from a table of SNPs from

Table 4: TaggingSNP: $UoI_{CUR}$, BasicCUR and GreedyCUR. See the text for details.

| Data | Size | $c$ | $UoI_{CUR}$ | BasicCUR | GreedyCUR |
|---|---|---|---|---|---|
| Yaledataset/SORCS3 | $1966 \times 53$ | 30 | 0.0096 | 0.0323 | 0.0062 |
| Yaledataset/PAH | $1979 \times 32$ | 20 | 0.0165 | 0.0308 | 0.0165 |
| Yaledataset/HOXB | $1953 \times 96$ | 36 | 0.0690 | 0.1369 | 0.0272 |
| Yaledataset/17q25 | $1962 \times 63$ | 35 | 0.0507 | 0.0895 | 0.0197 |
| HapMap/SORCS3 | $268 \times 307$ | 83 | 0.0023 | 0.0624 | 0.0023 |
| HapMap/PAH | $266 \times 88$ | 42 | 0.0087 | 0.0130 | 0.0053 |
| HapMap/HOXB | $269 \times 571$ | 57 | 0.0840 | 0.1696 | 0.0211 |
| HapMap/17q25 | $265 \times 370$ | 80 | 0.0421 | 0.1819 | 0.0162 |

different populations that capture the spectral information (or variations) of population. The subset of SNPs is called *tagging SNPs* (tSNPs). Here, we show how the $UoI_{CUR}$ method can be applied in this application to select columns (and thus tSNPs from the table of SNPs) which characterize the extent to which major patterns of variation of the intrapopulation data are captured by a small number of tSNPs.

We use the same two datasets used in [35], namely the Yale dataset and the Hapmap datset. The Yale dataset[1] [34] contains a total of 248 SNPs from around 2000 unrelated individuals from 38 populations each (from around the world). We consider four genomic regions (*SORCS3,PAH,HOXB, and 17q25*). The HapMap project[2] [22] (phase I) has released a public database of 1,000,000 SNP typed in different populations. From this database, we consider the data for the same four regions. Using the SNP table, an encoding matrix $A$ is formed with entries $\{-1, 0, 1\}$ indicating whether the expression exists ($\pm 1$) or not (0), with the sign indicating the order of the sequence. See supplementary material of [35] for details on this encoding. We obtained such encoded matrices from `http://www.asifj.org/`, as made available online by the authors of [35].

Table 4 lists the errors obtained from the three different methods, namely, $UoI_{CUR}$, BasicCUR and GreedyCUR [35] for different populations. The error reported is given by $nnz(\hat{A} - A)/nnz(A)$, where $A$ is the input encoding matrix, $C$ is the sampled/coarsened matrix, $\hat{A} = CC^{\dagger}A$, is the projection of $A$ onto $C$ and $nnz(A)$ is the number of elements in $A$. The GreedyCUR algorithm considers each column of the matrix sequentially, projects the remaining columns onto the considered column, and chooses the column that gave least error (where error is defined above). The algorithm then repeats the procedure to select the next column, and so on. This algorithm is very expensive, but it performs very well in practice. We observe that the $UoI_{CUR}$ algorithm performs better than

BasicCUR, and the performance is comparable with the unscalable GreedyCUR algorithm in many cases.

## A.10  Additional Discussion

It is common in many machine learning and data analysis methods to have either an implicit or explicit tradeoff between interpretability and prediction accuracy [2]. For example, in the context of unsupervised dimensionality reduction, CUR decompositions are low-rank approximations that are expressed in terms of a small number of actual rows and columns of the data matrix (i.e., actual data elements). It is known that they are provably only slightly worse in terms of variance reconstruction than the eigenvectors provided by PCA, but since they correspond to actual data elements, they are more easily interpretable in terms of the biological processes generating the data [30]. In a similar manner, in the context of supervised learning, UoI selects features and estimates parameters to optimize prediction accuracy while maintaining parsimony, resulting in interpretable models without substantially sacrificing prediction accuracy. Looking forward, the basic UoI framework can be applied to other algorithms to explore this trade-off more generally.

From a statistical perspective, we have illustrated four key properties of $UoI_{Lasso}$: control of false positive and false negative selection errors, and improved model selection consistency and data prediction accuracy. On both real and synthetic data, we observed generally improved prediction accuracy on held-out data, despite fewer parameters, a phenomenon we attribute to better up-front model selection reducing over-fitting on the training data. Importantly, when the feature space is dense, $UoI_{Lasso}$ has no systemic disadvantage relative to other methods (nor does it offer any advantage). It is common in many scientific fields to calculate a "score" (such as False Discovery Rate, FDR), independently for each feature, and then select features that exceed a (statistical) threshold, e.g., a p-value. This approach has several potential disadvantages. In particular, because of the independent (i.e., pair-wise) nature of some of these analysis, it is not possible to disambiguate features that uniquely contribute to a response, as opposed to simply co-vary with the causal features. Dependence between features can severely challenge most existing methods. Furthermore, even if estimation is done across all features simultaneously, selection is often done after estimation, and not in the context of response prediction (e.g., debiased Lasso). Additionally, the selection of a threshold (e.g., $p < 0.05$, multiple comparisons corrected) is often *a priori* arbitrary with respect to response prediction. Likewise, in the generation of genetic and brain networks, the underlying graphs need to be estimated from data. In practice, experimenters often apply a *post hoc* threshold to the distribution of edge weights (often, marginal correlation) outside the context of optimizing prediction accuracy (the objective function). These *ad hoc* processes challenge rigorous mathematical analysis; they can dramatically alter the structure of empirical results; and they are likely a major source of error in downstream scientific conclusions. In contrast, using the UoI approach for model selection

and model estimation to optimize prediction suggests a potentially general, normative framework for overcoming these and other related algorithmic-statistical issues in a scalable way.

From an algorithmic perspective, $UoI_{Lasso}$ efficiently constructs a family of model supports by combining randomized data resampling with a range of regularization hyperparameters, imbuing $UoI_{Lasso}$ with a high degree of parallelization, making it a natural fit for modern distributed computing platforms. We provide open-source implementations of $UoI_{Lasso}$ in Matlab, R, and Python to make it as broadly and easily accessible as possible. Additionally, we provide versions of $UoI_{Lasso}$ in Python that use either OpenMP or Spark for efficient implementation on a variety of distributed computing platforms [20]. The current computing bottleneck in $UoI_{Lasso}$ for application to massive data sets is the calculations involved in solving the core Lasso/OLS problem, suggesting that recent work in distributed convex optimization or sampling-based techniques from randomized linear algebra may lead to still further benefits.

From the perspective of scalable, interpretable, scientific data analysis, the modular structure of UoI is particularly powerful, as it allows for a diversity of methods to be used, making it both general and flexible. In the context of linear regression, use of BoLasso for model selection and ordinary least squares (OLS) for model estimation resulted in the $UoI_{Lasso}$ algorithm primarily studied here; but the UoI framework can accommodate other base methods such as stability selection [33], SCAD [18], debiased Lasso [27], or other specialized, problem specific methods. Alternatively, in the context of classification, support vector machines or other classifiers could have been used [13]. More generally, we see no reason why the UoI framework could not be extended to more complex statistical models, such as random forests [11], auto-regressive models, and canonical correlation analysis [26]. Relatedly, while data analysis methods in science are often heavily tailored to a specific domain, the UoI method is a modality-agnostic data analytic method, and the problems for which we have demonstrated its utility (regression and classification) are ubiquitous across data domains in science and industry, e.g., material science and climate science, in addition to neuroscience and genetics. Therefore, there is every reason to believe that UoI and related methods could enhance interpretable scientific machine learning in other scientific fields in gigabyte, terabyte, and petabyte sized data sets that are increasingly common.