[Reviews · NeurIPS 2017]

Reviewer 1



The paper proposes a new model selection and estimation method for problems with underlying sparsity. The selection stage involves intersections of model support resulting from several bootstrap samples. Estimation involves bagging estimate of best models corresponding to different regularization parameters. Overall the model is both intuitively appealing, and shows good performance in important applications (eg. estimating complex phenotype from genes). Would recommend for publication.

Reviewer 2



This paper focuses on model selection and, to some extent, feature selection in large datasets with many features, of which only a small subset are assumed to be necessary for accurate prediction. The authors propose a general method by which model selection is performed by way of feature compression performed by taking the intersection of a multiple regularization parameters in an ensemble method, and then model estimation by taking a union over multiple outputs. (This is also the major contribution of the paper.) A second contribution is found in the union operation in a model averaging step with a boosting/bagging flavor. Overall, I found the paper's method section well written and the idea proposed to be complete. The paper's experimental section was difficult to follow, but the results do seem to support the framework. One major missing part of the paper is a reasonable discussion of using the framework beyond a Lasso base. Are there reasons why this method would not work for classification? Are there potential hitches to using this method with already-ensemble-based methods like random forests? While there are many uses for the UoI with a Lasso base already, it was increase the general interest of the framework if UoI could be placed in the more general ML space.

Reviewer 3



This paper proposes a general framework for interpretable prediction. The authors consider a two-stage (intersection + union) approach to achieve the goal of guaranteeing both interpretability(sparsity) and accuracy. The authors applied the method to synthetic datasets and some biomedical datasets. The paper is well written with good demonstrations and easy to understand. The main algorithm/framework is not well explained in the main context. Theorem 1 for UoI_Lasso in the appendix should be presented more rigorously by listing the assumptions and the exact formulas for constants C and C_1. The authors claimed the union of intersections as the major innovation. However, this approach lacks novelty since similar techniques are widely used in the data mining community such as different forms of filter + wrapper + embedded methods. The UoI can be viewed as a special case of forward selection + backward elimination. The extensive experiments are convincing. But the authors should compare with existing methods with forward selection + backward elimination structures.